# Direct electrosynthesis and separation of ammonia and chlorine from waste streams via a stacked membrane-free electrolyzer

Jianan Gao [1], Qingquan Ma [1], Zhiwei Wang [2], Bruce E. Rittmann [3] & Wen Zhang [1,4] ✉

Electrosynthesis, a viable path to decarbonize the chemical industry, has been harnessed to generate valuable chemicals under ambient conditions. Here, we present a membrane-free flow electrolyzer for paired electrocatalytic upcycling of nitrate ($NO_3^-$) and chloride ($Cl^-$) to ammonia ($NH_3$) and chlorine ($Cl_2$) gases by utilizing waste streams as substitutes for traditional electrolytes. The electrolyzer concurrently couples electrosynthesis and gaseous-product separation, which minimizes the undesired redox reaction between $NH_3$ and $Cl_2$ and thus prevents products loss. Using a three-stacked-modules electrolyzer system, we efficiently processed a reverse osmosis retentate waste stream. This yielded high concentrations of $(NH_4)_2SO_4$ (83.8 mM) and NaClO (243.4 mM) at an electrical cost of 7.1 kWh per kilogram of solid products, while residual $NH_3/NH_4^+$ (0.3 mM), $NO_2^-$ (0.2 mM), and $Cl_2/HClO/ClO^-$ (0.1 mM) pollutants in the waste stream could meet the wastewater discharge regulations for nitrogen- and chlorine-species. This study underscores the value of pairing appropriate half-reactions, utilizing waste streams to replace traditional electrolytes, and merging product synthesis with separation to refine electrosynthesis platforms.

Conventional chemical industries depend on fossil fuels and emit large amounts of greenhouse gases[1]. In contrast, electrosynthesis is an emerging redox platform that can achieve more environmentally compatible chemical production that is more amenable to using renewable energy sources (i.e., solar and wind)[2,3]. For example, renewable feedstock -- air, water, $CO_2$, and derivatives of biomass -- have been converted to portable fuels, such as $NH_3$ and $C_2H_5OH$, and to important industrial chemicals, such as $Cl_2$, CO, $H_2$, $C_2H_4$, and $CH_3OH$[2].

Today, electrolytes such as tetrahydrofuran, toluene, and inorganic salts are employed to enhance electron transfer in electrosynthesis processes[3-8]. These inputs increase the costs of input materials and for treating secondary wastes. Conversely, wastewaters that contain dissolved contaminants could be utilized as electrolytes, offering a globally abundant and underexploited resource to tap into[9-11]. Approximately $2.2 \times 10^{15}$ L of wastewater, constituting 54% of total freshwater withdrawals, is generated annually across municipal, agricultural, and industrial sectors[12]. For instance, the electrocatalytic valorization of chlorinated organic water pollutants to ethene was recently proven feasible[13]. Minimizing the costs of input materials, avoiding secondary contaminants, and electrochemically valorizing waste elements will offset wastewater treatment costs[14,15].

An important example is the conversion of $NO_3^-$ and $Cl^-$ ions to $NH_3$ and $Cl_2$ gases, which are chemicals produced globally at approximately 182 million and 88 million metric tons per year, respectively[16-20]. $NO_3^-$ and $Cl^-$ are commonly present in industrial

[1]Department of Civil and Environmental Engineering, New Jersey Institute of Technology, Newark, NJ, US. [2]State Key Laboratory of Pollution Control and Resource Reuse, Shanghai Institute of Pollution Control and Ecological Security, Tongji Advanced Membrane Technology Center, School of Environmental Science and Engineering, Tongji University, Shanghai, China. [3]Biodesign Swette Center for Environmental Biotechnology, Arizona State University, Tempe, AZ, US. [4]Department of Chemical & Materials Engineering, New Jersey Institute of Technology, Newark, NJ, US. ✉e-mail: wen.zhang@njit.edu

wastewater, such as ion-exchange brines, which may contain 150 mM $NO_3^-$ and 5 wt% NaCl[21,22]. Electrocatalytic conversion of nitrate to ammonia, which has been demonstrated[14,23], involves cathodic nitrate reduction (Eq. 1) coupled to an anodic reaction such as water oxidation (Eq. 2)[9,24]. Similarly, industrial chlorine gas ($Cl_2$) is primarily produced by the chlor-alkali process, which consists of an anodic chlorine-evolution reaction paired with a hydrogen ($H_2$)-evolution reaction (Eqs. 3 and 4)[25,26].

$$Cathode-1: \quad NO_3^- + 6H_2O + 8e^- \rightarrow NH_3 + 9OH^- \quad (1)$$

$$Anode-1: \quad 2H_2O \rightarrow O_2 + 4H^+ + 4e^- \quad (2)$$

$$Anode-2: \quad 2Cl^- \rightarrow Cl_2 + 2e^- \quad (3)$$

$$Cathode-2: \quad 2H_2O + 2e^- \rightarrow H_2 + 2OH^- \quad (4)$$

While today's processes separately generate $O_2$ and $H_2$, it makes sense to couple the nitrate-to-ammonia conversion with chlorine evolution. To synchronize $NH_3/Cl_2$ production, the rapid reaction between $NH_3$ and $Cl_2$ (a rate constant of *c.a.* $4.2 \times 10^6 M^{-1} \cdot s^{-1}$) must be prevented[27]. One approach involves the use of ion-selective or -exchange membranes to separate the cathode and anode and their respective $NH_3$ and $Cl_2$ productions[28–30]. However, the substantial initial cost of membrane material costs—~ 24% of the electrolyzer-stack costs—and problems related to the durability and requisite maintenance of the membranes lead to high capital and operating costs of the electrolyzer[31–33]. Therefore, it would be of value to devise a process free from ion selective/exchange membrane for the synchronous production and extraction of $NH_3/Cl_2$ products, as long as product purity, product yield rate, and efficiency are optimally balanced.

Recently, membrane modules integrated with hydrophobic gas-diffusion layers have emerged as effective tools for gaseous compound extraction (e.g., $CH_4$ and $H_2$)[34], delivery of $CO_2$ and $N_2$[35], and hybrid processes. When these hydrophobic interfaces operate below their liquid entry pressure, they establish a triphasic boundary, working as a liquid water barrier, but allowing the passage of gases. In addition, by integrating another electrocatalyst layer into the membrane module and referencing Fick's law, we note that Faradaic reactions involving proton consumption or production result in localized pH extremes: alkaline conditions (>11.5) at the cathode and acidic conditions (<2.5) at the anode, even with modest current densities of $5 mA \cdot cm^{-2}$ (Supplementary Fig. 1). Such pH environments can promote the formation of gaseous $NH_3$ and $Cl_2$ at their respective primary interfaces[16,36]. By directing separation at the membrane-water junction instead of at the traditional gas-liquid interface, we hypothesize that bespoke chemical reactions on the membrane surface might achieve the production and separation of $NH_3/Cl_2$ products while simultaneously minimizing product losses.

In this report, we present an electrified membrane-free electrolyzer featuring gas-extraction electrodes for synchronous $NH_3/Cl_2$ production and extraction (Fig. 1a). First, we demonstrated that the electrode assembly combining electrocatalyst layer and gas exchange layer can effectively balance the production and separation of $NH_3$ and $Cl_2$. Building on this, we integrated the gas-extraction electrodes into a flow-type, membrane-free electrolyzer, achieving synchronous electrosynthesis and separation of $NH_3$ and $Cl_2$ with high product purity, high yield rates, and minimal product loss. Our comprehensive investigation delves into the electrochemical conversion pathways, the homogeneous redox dynamics of nitrogen- and chlorine-derived species, and the mechanisms facilitating the selective extraction of $NH_3$ and $Cl_2$, thereby enriching our understanding of the complex interactions within the system. We then successfully implemented a stacked electrolyzer comprised of three modules and a geometric electrode area of up to 300 cm². This system efficiently processed the actual reverse osmosis retentate waste stream, resulting in high product concentrations (($NH_4)_2SO_4$: 83.8 mM, NaClO: 243.4 mM) and low residual intermediates/products ($NH_3/NH_4^+$: 0.3 mM, $NO_2^-$: 0.2 mM, $Cl_2/HClO/ClO^-$: 0.1 mM). This research not only opens avenues for the upscaling of electrosynthesis platforms using waste streams in place of traditional electrolytes but also provides critical insights into product synthesis and separation pathways.

## Results

### Electrode assembly design and basic performance evaluation

To achieve synergistic electrosynthesis and separation of $NH_3$ and $Cl_2$ from waste streams, the key is to develop electrode assemblies with high catalytic activity and gas-transfer rate. Metallic copper and ruthenium oxide were selected as model electrocatalysts for nitrate reduction reaction ($NO_3RR$) and chlorine evolution reaction (CER) due to their rapid reduction/oxidation kinetics of $NO_3^-$ and $Cl^-$ towards $NH_3$ and $Cl_2$[22,37]. The electrocatalysts (hierarchical Cu or $RuO_2$ particles) were further immobilized to a carbon-polytetrafluoroethylene (PTFE)-based gas diffusion layer to obtain a gas-extraction electrode (Supplementary Fig. 2). Scanning electron microscopy (SEM) demonstrated the uniform loading of the electrocatalytic layer (Supplementary Fig. 3), while the X-ray diffraction (XRD) patterns confirmed the successful fabrication of Cu or $RuO_2$ dominated electrocatalytic layer (Supplementary Fig. 4).

We then examined separately the electrosynthesis and separation rates of the gas extraction electrodes for $NH_3$ and $Cl_2$. A synthetic medium-strength waste stream containing either 25 mM $NO_3^-$ or 25 mM $Cl^-$ (pH = 7.0 ± 0.1) was employed. We posited that spontaneous stripping of $NH_3$ and $Cl_2$ would occur at the gas diffusion layer, propelled by a concentration gradient in local vapor pressure across the electrode module (Fig. 1a). As depicted in Fig. 1b, increasing the cathodic potential from − 0.50 to − 0.80 V vs. RHE corresponded to a surge in the $NH_3$ yield rate, from 9.8 ± 1.1 to 49.4 ± 0.7 × 10⁻¹⁰ $M \cdot NH_3 \cdot cm^{-2} \cdot s^{-1}$, accompanied by an upswing in the $NH_3$ transfer rate. Beyond − 0.80 V vs. RHE, the $NH_3$ yields stabilized between 49.1 ± 2.7 to 50.1 ± 2.2 × 10⁻¹⁰ $M \cdot NH_3 \cdot cm^{-2} \cdot s^{-1}$, but its transfer rate kept diminishing. This phenomenon can be ascribed to the enhanced Faradaic Efficiency (FE) for $H_2$ production at elevated cathodic potentials (Supplementary Fig. 5). The concurrent efflux of $H_2$ competes for the gas transfer channels with $NH_3$, resulting in a diminished $NH_3$ transfer rate. Meanwhile, for $Cl_2$, Fig. 1c illustrates an S-shaped relationship with applied anodic potential, where $Cl_2$ yield and transfer rates fluctuated between 34.9 ± 1.3 to 35.2 ± 1.7 × 10⁻¹⁰ $M \cdot Cl_2 \cdot cm^{-2} \cdot s^{-1}$) was reached when the anode potential was greater than 2.45 V vs. RHE, beyond which the rate was constrained by the limited mass transfer of $Cl^-$. Our findings underscore that $NH_3$ and $Cl_2$ yields were influenced by the applied potential, which also controlled the product transfer rates. Furthermore, the $NH_3$ separation efficiency (defined as the ratio of the molar amount of separated $NH_3$ to the total molar amount of $NH_3$ produced) had a volcano-shaped response to the cathodic potential, peaking at −0.80 V vs RHE with 90 ± 2% efficiency (Fig. 1d). In contrast, the $Cl_2$ separation efficiency performed stable with an average value of 99 ± 1% across a broad potential range (Fig. 1e). Thus, when the $NO_3RR$ and CER reactions were synchronized, anodic potential could be used to sensitively match the cathode potential. The high product-separation efficiency hints at the feasibility of a unified $NH_3$ and $Cl_2$ electrosynthesis-separation in one membrane-free electrolyzer.

### The synchronous electrosynthesis and separation of $NH_3$ and $Cl_2$

We then investigated the performance of simultaneous $NH_3$ and $Cl_2$ electrosynthesis and separation. The gas-extraction electrodes were incorporated into a flow-type membrane-free electrolyzer, which consisted of an ammonia trap channel (circulating pH 1.0 ± 0.1 $H_2SO_4$

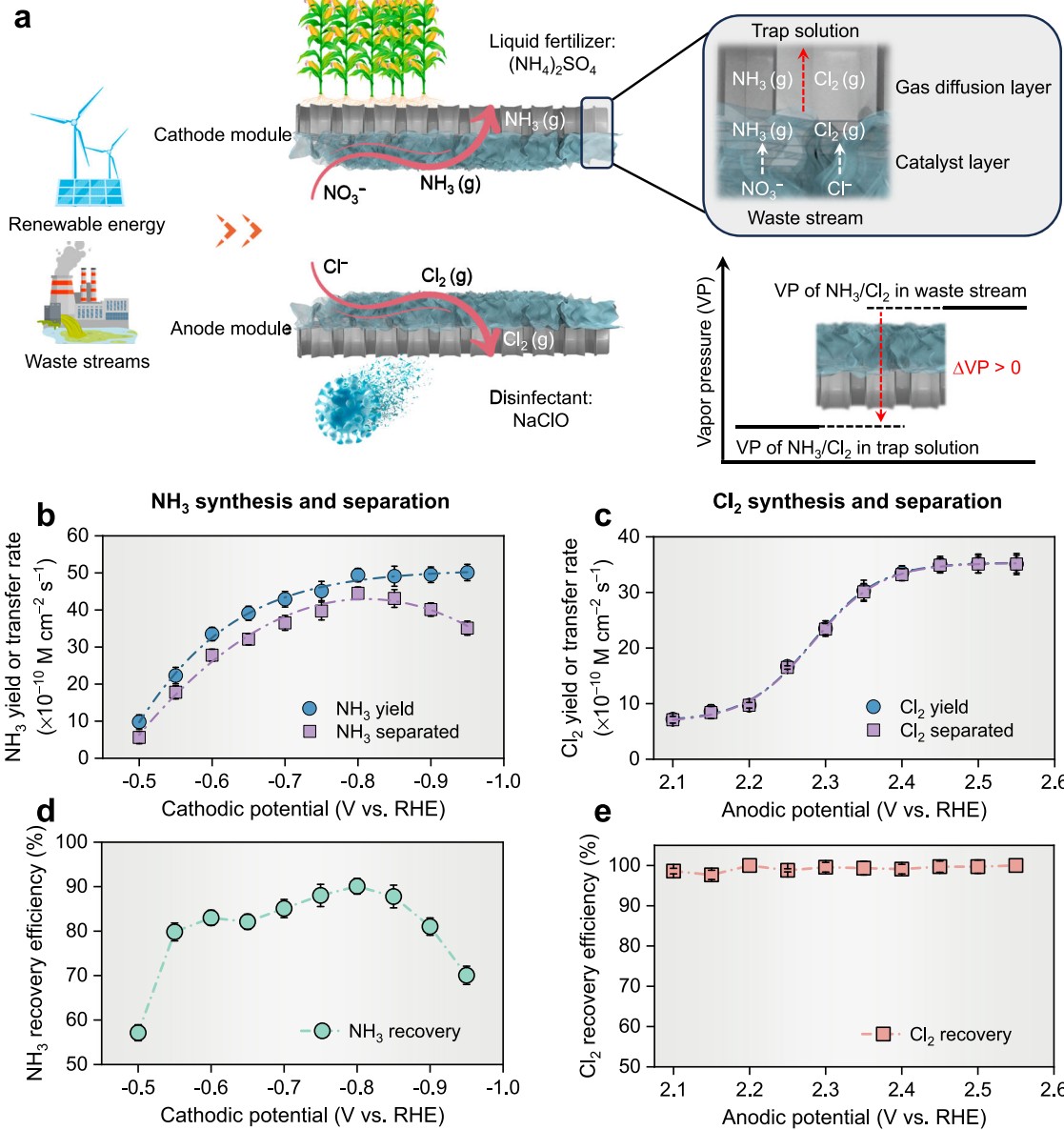

**Fig. 1 | Concept and verification of synergistic electrosynthesis and separation of NH₃ and Cl₂. a** Schematic of the electrochemical NH₃ and Cl₂ production under ambient conditions using renewable energy and waste stream. **b, c** The yield and separation rates of NH₃ and Cl₂ as a function of applied potentials on cathodic and anodic electrode assemblies, respectively. **d, e** The recovery efficiencies of yield NH₃ and Cl₂ as a function of applied potentials on cathodic and anodic electrode assemblies, respectively. No iR compensation was applied. The synthetic waste stream is 25 mM NO₃⁻ or 25 mM Cl⁻ mixed with 0.1 M Na₂SO₄ (pH = 7.0 ± 0.1) to simulate co-existing ions in the waste stream. The error bars represent the standard deviations from triplicate tests.

solution), a chlorine trap channel (circulating pH 13.0 ± 0.1 NaOH solution), and a waste stream channel (Fig. 2a). The physical installation diagram of the electrolyzer is shown in Supplementary Fig. 6. The gas extraction electrodes separated two trap channels from the middle waste stream channel. Nitrate and chlorite underwent interfacial electrochemical reactions at the electrodes and were converted into gaseous products (NH₃ and Cl₂) with synchronous transfer across the gas diffusion layers into the trap electrolytes. We hypothesized that rapid extraction rates for NH₃ and Cl₂ could obviate their contact within the sewage stream, which would effectively thwart the undesirable reactions between NH₃/NH₄⁺ and reactive chlorine species such as HClO/ClO⁻ in the electrolyte, leading to N₂ and Cl⁻ as final products. The stripped NH₃ and Cl₂ were chemically converted to (NH₄)₂SO₄ and NaClO, respectively, within their designated trap channels.

Based on potential-controlled experiments, various constant cell potentials were utilized, in contrast to employing individually controlled cathodic or anodic potentials, to balance the NH₃ and Cl₂ production and separations. Initially, we determined the baseline concentrations of NH₃ and Cl₂ at this cell potential without the interactions between products. As demonstrated in Fig. 2b, c, after 5 h of single-electrosynthesis with the synthetic waste stream of either 25 mM NO₃⁻ or 25 mM Cl⁻, the concentrations of NH₃ and Cl₂ in their respective trap solutions reached 17.9 ± 0.5 mM and 9.4 ± 0.2 mM. Subsequently, we introduced a synthetic mixed waste stream containing 25 mM NO₃⁻ and 25 mM Cl⁻ to monitor the product concentrations during co-electrosynthesis of NH₃ and Cl₂. The typical I-t curve and pH variations were illustrated in Supplementary Fig. 7, confirming the stability of the system. As shown in Fig. 2d, the final NH₃ and Cl₂ concentrations in the trap solutions were 17.1 ± 0.2 mM and 9.3 ± 0.4 mM, respectively, which were close to their baseline values. Notably, the final NH₃ concentration in the waste stream during co-electrosynthesis (1.6 ± 0.1 mM) was lower than that during single-

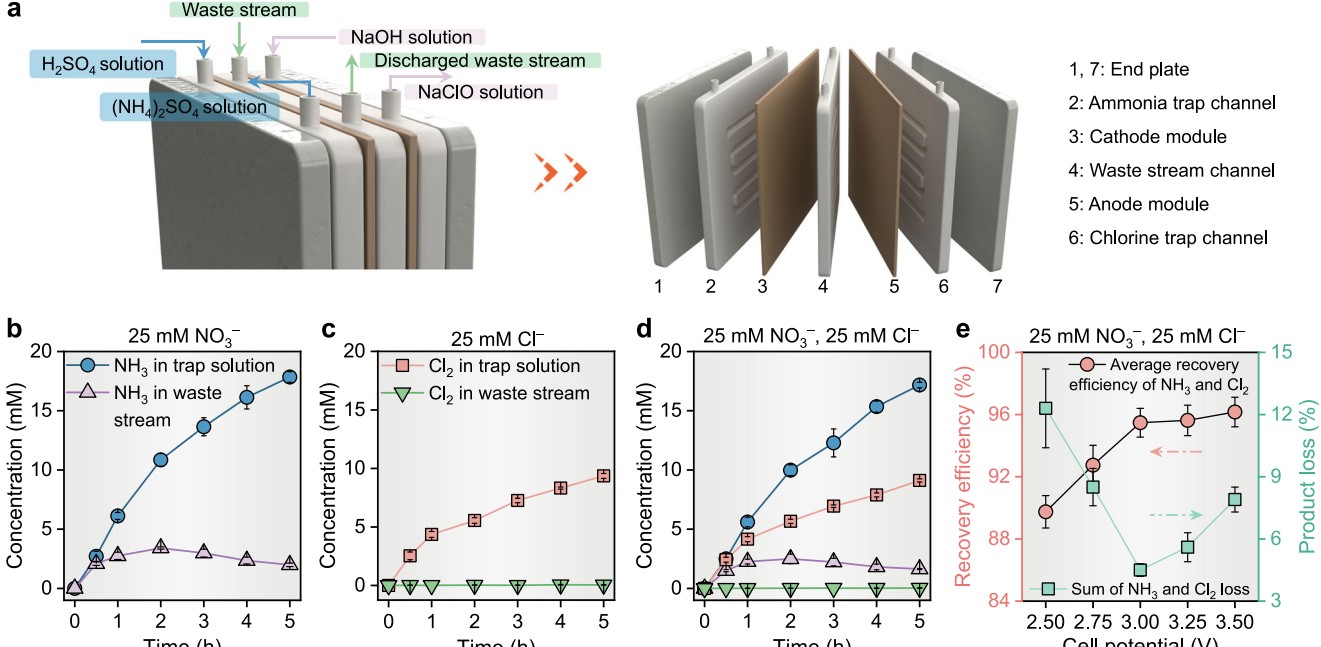

**Fig. 2 | The performance of synchronous electrosynthesis and separation of NH₃ and Cl₂ of the flow-type membrane-free electrolyzer. a** Schematics and configuration of this flow-type membrane-free electrolyzer for electrochemical synthesis and in situ recovery of ammonium sulfate and hypochlorous acid from waste streams. **b, c** Single-electrosynthesis for NH₃ (**b**) and Cl₂ (**c**): the concentrations of NH₃ or Cl₂ in the trap solution and the waste stream as a function of reaction time by feeding 25 mM NO₃⁻ or 25 mM Cl⁻ mixed with 0.1 M Na₂SO₄ (pH = 7.0 ± 0.1). **d** Co-electrosynthesis for NH₃ and Cl₂: the concentrations of NH₃ and Cl₂ in the trap solution and the waste stream as a function of reaction time by feeding 25 mM NO₃⁻ and 25 mM Cl⁻ mixed with 0.1 M Na₂SO₄ (pH = 7.0 ± 0.1). **e** The average recovery efficiencies and the sum of the product loss of NH₃ and Cl₂ at different total cell potentials. The error bars represent the standard deviations from triplicate tests.

electrosynthesis (2.0 ± 0.2 mM), while Cl₂ was even undetectable in the waste stream for both processes. This reduced NH₃ concentration in the waste stream could be attributed to the oxidation of NH₃/NH₄⁺ by Cl₂/HClO/ClO⁻ species. When the cell potential increased from 2.5 V to 3.5 V, the average recovery efficiency (pink data points) for NH₃ and Cl₂ improved from 90 ± 1% to 96 ± 1%, as Fig. 2e and Supplementary Fig. 8 indicate. The total loss of produced NH₃ and Cl₂ (green data points) varied between 12 ± 2% to 5 ± 1%, with the minimum value located at 3.0 V. These results validate the membrane-free electrolyzer's effectiveness in co-electrosynthesizing NH₃ and Cl₂ with high efficiency and acceptable product loss. Furthermore, the interaction between residual nitrogen and chloride species in the waste stream resulted in the formation of N₂ and Cl⁻ as final products, further minimizing the residual products such as NH₃/NH₄⁺ and Cl₂/HClO/ClO⁻.

### Probing the mechanism of NH₃/Cl₂ separation on reducing product loss

To gain insights into the effect of NH₃/Cl₂ separation on reducing product loss, we conducted a series of control experiments by varying the concentrations/ratios of nitrogen and chloride species in the feed electrolyte, with and without the incorporation of separation operations. The major heterogeneous and homogeneous redox reactions within the electrolyzer are shown in Fig. 3a and summarized in Supplementary Table 1[38]. As shown in Fig. 3b, c, the introduction of Cl⁻ ions resulted in an observable increase in the remaining NO₃⁻ concentration (blue data points), from 1.4 ± 0.3 mM to 9.0 ± 0.3 mM, and a corresponding decrease in the final NH₃ (purple data points) concentration, from 13.1 ± 0.2 mM to 1.7 ± 0.2 mM. The concentration of N₂ (green data points), encompassing both dissolved and vaporized forms, increased from 9.6 ± 0.3 mM to 14.3 ± 0.1 mM. This calculation was based on the disparity between the input nitrate nitrogen and the nitrogen species retained in the solution. After 4 h, the stability of NH₃ concentration, despite a decreasing NO₃⁻ concentration, is attributed

to the direct oxidation of NH₃ under alkaline conditions[38], as indicated by the electrolyte pH rise above 11.5. The amplified N₂ concentration when introducing Cl⁻ is attributed to the more rapid reaction kinetics toward N₂ ($4.2 \times 10^6 \, M^{-1} \cdot s^{-1}$) compared to NO₃⁻ ($0.1 - 0.7 \, M^{-1} \cdot s^{-1}$) in the context of NH₃/NH₄⁺ interaction with HClO/ClO⁻[27,38]. As the rate-limiting species for NO₃⁻ reduction, the average NO₂⁻ concentration within 10 h electrolysis (gray data points) reduced from 1.9 ± 0.1 mM to 0.9 ± 0.1 mM, when Cl⁻ was present.

Subsequent experiments aimed to trace the NO₂⁻ conversion pathways (e.g., conversion to NO₃⁻ or NH₃) and to ascertain the oxidation priorities of active chlorine species with NO₂⁻ and NH₃. To this end, equal amounts of NO₂⁻ and NH₃ were introduced together to the electrolyte, and the evolution of subsequent nitrogen species was monitored. Figure 3d indicates that during the initial 3 h, the NO₂⁻ concentration decreased from 12.5 mM to 1.2 ± 0.2 mM and was mainly oxidized to NO₃⁻ that increased from 0 mM to 8.3 ± 0.2 mM. Meanwhile, the N₂ and NH₃ concentrations increased from 0 mM and 12.5 mM to 1.1 ± 0.2 mM and 14.4 ± 0.5 mM, respectively. After 3-h, when NO₂⁻ was nearly depleted, NH₃ oxidation became dominant, as indicated by the reduced NH₃ concentration from 14.4 ± 0.4 mM to 2.2 ± 0.2 mM and the increased N₂ concentration from 1.1 ± 0.2 mM to 13.4 ± 0.4 mM. From a reaction kinetic standpoint, NH₃ has multi-step conversions with rate constants spanning from $1.7 \times 10^2$ to $3.1 \times 10^6$ $M^{-1} \cdot s^{-1}$ and is considerably more vulnerable to HOCl-induced oxidation than NO₂⁻ that has multi-step conversions with measured rate constant for only one step by far ($1.8 \times 10^5 \, M^{-1} \cdot s^{-1}$)[39–42]. The preferential reaction of NO₂⁻ with HOCl can be explained by the breakpoint chlorination mechanism in NH₃, which occurs when the HOCl to NH₃ mole ratio gradually reaches 1.5[38]. The continuous consumption of HOCl by NO₂⁻ prevents the system from reaching this breakpoint chlorination ratio. During HOCl-mediated NH₃ oxidation to N₂ and NO₃⁻, chloramine intermediates react with NO₂⁻, converting back to NH₃ as end products[42]. This process further influences the dynamics of

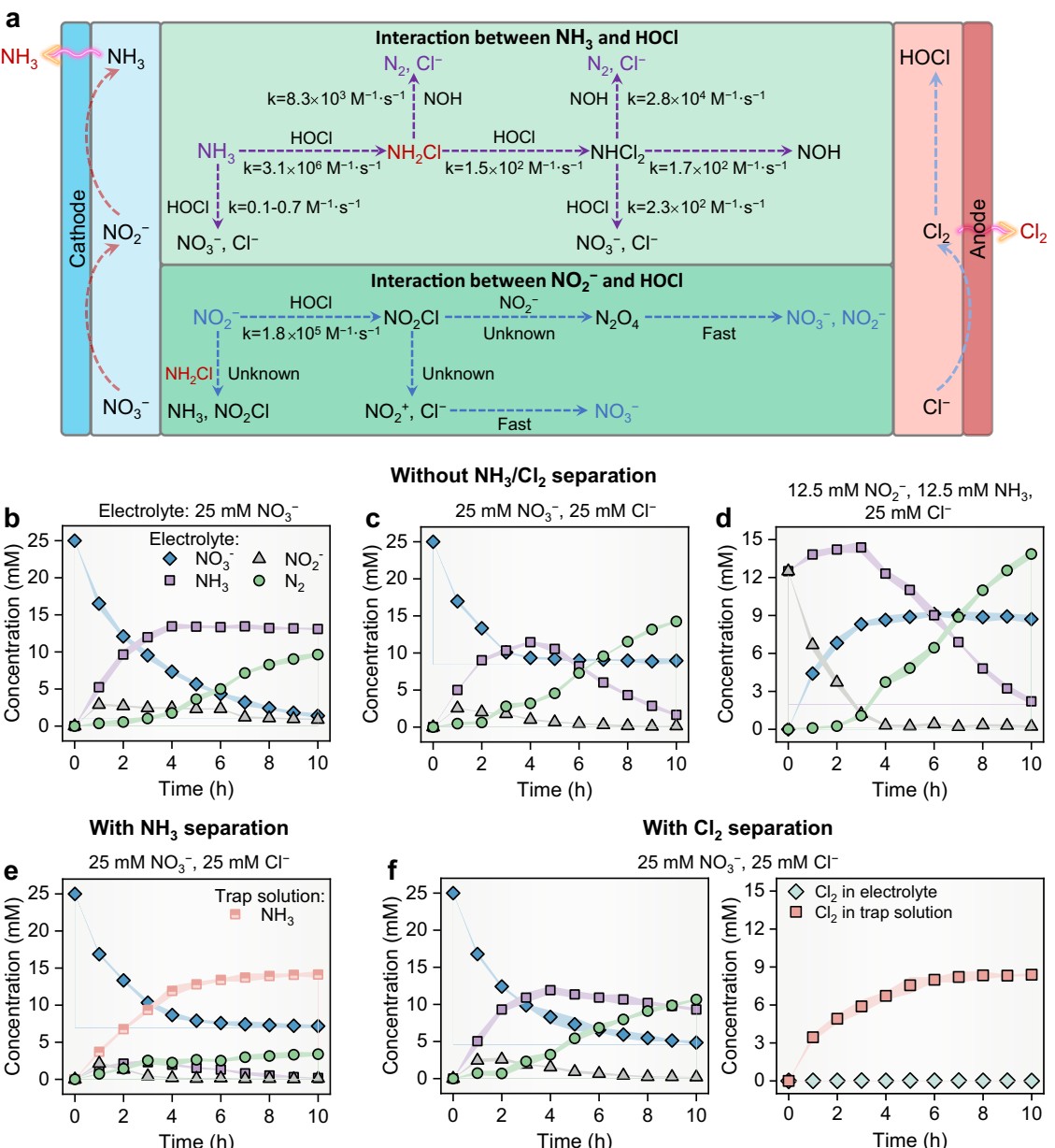

**Fig. 3 | Mechanism analysis. a** The major heterogeneous and homogeneous redox reactions within the electrolyzer. **b**–**f** the nitrogen species and active chlorine species evolution over reaction time when the electrolytes in the reaction chamber contained 0.1 M Na$_2$SO$_4$ mixed with (**b**) 25 mM NO$_3^-$; (**c**) 25 mM NO$_3^-$, 25 mM Cl$^-$; (**d**) 12.5 mM NO$_2^-$, 12.5 mM NH$_3$, 25 mM Cl$^-$, pH = 12.0 ± 0.1; (**e**) 25 mM NO$_3^-$, 25 mM Cl$^-$; (**f**) 25 mM NO$_3^-$, 25 mM Cl$^-$. All experiments were carried out under a total cell potential of 3.0 V. The shadow area in the figure represents the error scale. The error scales represent the standard deviations from triplicate tests.

NH$_3$/HOCl interactions. The main interference arises from HOCl that converts NO$_2^-$ to NO$_3^-$ at a kinetic rate five orders of magnitude faster than chloramine reactions due to the low HOCl concentration condition within the electrolyzer[43].

We further assessed the influence of the separation of NH$_3$ and Cl$_2$ from the electrolyte on the final product formation. Attaching an ammonia trap channel next to the gas-permeable cathode (Fig. 3e) separated over 99% of the produced NH$_3$ from the electrolyte channel (red data points). Consequently, the NO$_3^-$-conversion efficiency increased by 11%, and nitrogen loss evidenced by N$_2$ formation also decreased by 76%. A similar experiment was conducted by incorporating a single chlorine trap channel (without the use of the ammonia trap channel), which yielded a modest improvement of the NO$_3^-$ conversion efficiency by 46% and the reduced N$_2$ loss by 25% according to the results in Fig. 3f. Although ~100% of the

generated Cl$_2$ was extracted from electrolyte, after 5 h electrolysis, the Cl$_2$ concentration in the chlorine trap solution (7.5 ± 0.5 mM) was still lower than that in Fig. 2c (9.1 ± 0.1 mM), where the separation of NH$_3$ and Cl$_2$ occurred simultaneously. This observation implies that the average Cl$_2$ recovery efficiency of 99% across a broad anodic potential range in Fig. 1e may be misleading. The Cl$_2$ extraction kinetics rate is not fast enough to efficiently separate all produced Cl$_2$ at the anodic interface, which consequently causes the residual HClO and ClO$^-$ in the waste stream. These residual species are likely to engage in side reactions with NH$_3$ and NO$_2^-$, resulting in the formation of Cl$^-$. Therefore, the optimal electrosynthesis and separation performance of the membrane-free electrolyzer required synchronous extraction processes for NH$_3$ and Cl$_2$. Isolating ammonia or chlorine gas alone was inadequate to mitigate product loss.

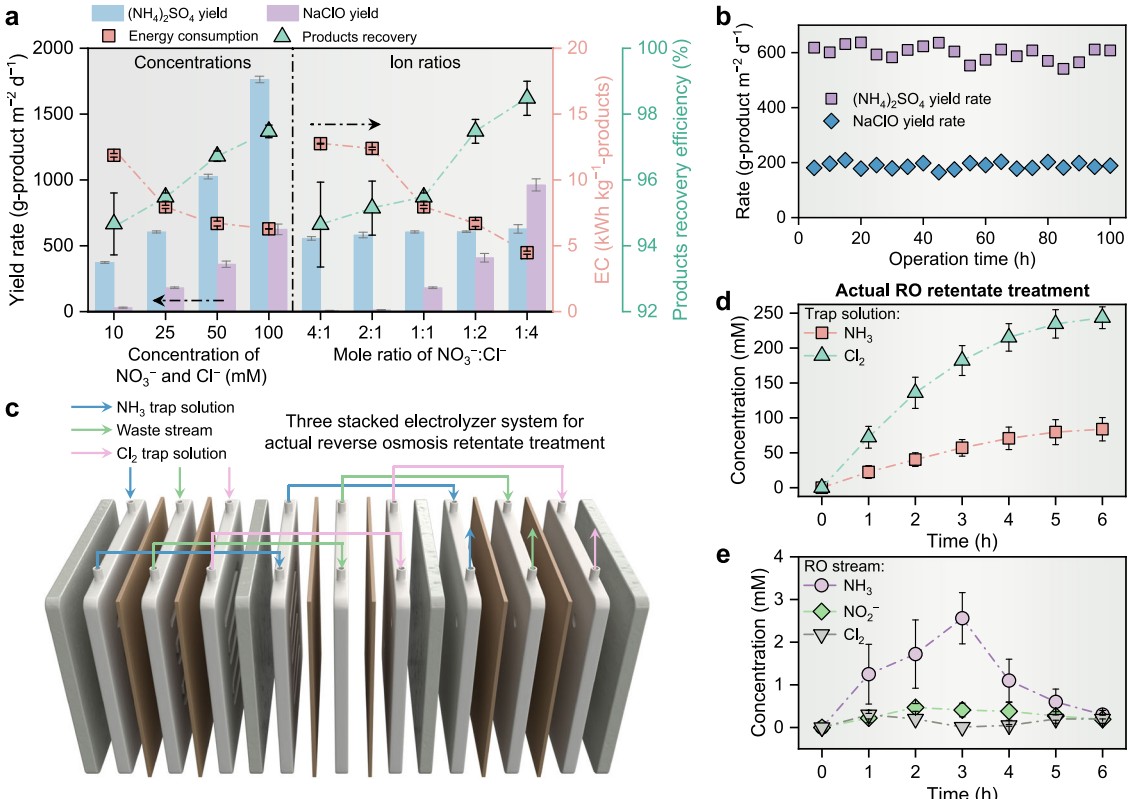

**Fig. 4 | Scalable electrosynthesis by using stacked flow-type membrane-free electrolyzer system. a** Comparison of products ((NH$_4$)$_2$SO$_4$ and NaClO) yield rate, energy consumption for production, and products recovery efficiency. Testing conditions are shown in the experimental section (i.e., pH = 7.0 ± 0.1, Na$_2$SO$_4$ = 0.1 M; NO$_3^-$ = 10–100 mM, Cl$^-$ = 6.25–100 mM, cell potential: 3.0 V). **b** Cycle performance of the membrane-free electrolyzer without changing the cathodic/anodic modules at 3.0 V total cell potential, pH = 7.0 ± 0.1, 0.1 M Na$_2$SO$_4$, 25 mM NO$_3^-$, 25 mM Cl$^-$. Each cycle undergoes 5 h with >85% NO$_3^-$ is converted to NH$_3$, and > 90%

NH$_3$ is separated from the electrolyte. **c** Illustration of stacked electrolyzer system consisting of three modules. **d, e** The performance of stacked electrolyzer system for actual reverse osmosis (RO) retentate treatment. Specifically, the concentration of recovery NH$_3$ and Cl$_2$ in trap solutions and residual concentrations of NH$_3$/NH$_4^+$, Cl$_2$/HClO/ClO$^-$, and NO$_2^-$ in the RO stream as a function of operation time under a total cell potential of 3.0 V were recorded. The error bars represent the standard deviations from triplicate tests.

## Large-scale electrosynthesis using a stacked electrolyzer system

The practical implementation and performance (e.g., productivity and product concentration, energy consumption, and intermediates/products residual) of this membrane-free electrolyzer are highly affected by complex water chemistry of the feeding waste stream (e.g., concentrations and ratios of NO$_3^-$ and Cl$^-$ ions), which should be considered. Supplementary Figs. 9 and 10 show the time-resolved evolution of nitrogen and chlorine species for synthetic waste streams with different nitrate and chloride concentrations or ratios. The pH of all synthetic waste streams was controlled to be 7.0 ± 0.1. The relevant current density and NH$_3$/Cl$_2$ FE are summarized in Supplementary Table 2.

The yield rate and energy consumption for the production of (NH$_4$)$_2$SO$_4$ and NaClO, along with NH$_3$/Cl$_2$ recovery efficiencies from the waste stream, were measured and summarized for the different feed-solution chemistries. We first increased the same molar concentrations of NO$_3^-$ and Cl$^-$ in the feed solution from 10 mM (typical for most industrial wastewater[44]) to 100 mM found in brine wastewater from ion exchange or reverse osmosis processes[21], under a fixed cell potential of 3.0 V. The left side of Fig. 4a shows the high concentration of NO$_3^-$ and Cl$^-$ led to a fast reaction kinetics of NO$_3^-$-to-NH$_3$ and Cl$^-$-to-Cl$_2$ conversions and thus decreased the overall energy consumption from 11.9 ± 0.1 to 6.3 ± 0.1 kWh·kg$^{-1}$-products, accompanied by the increased NH$_3$/Cl$_2$ average recovery efficiency (95 ± 1% to 97 ± 0%) and two product generation rates or fluxes (373 ± 7 to 1763 ± 25 g-(NH$_4$)$_2$SO$_4$·m$^{-2}$·d$^{-1}$ and 30 ± 3 to 625 ± 14 g-NaClO·m$^{-2}$·d$^{-1}$, respectively). This result indicates that the NH$_3$/Cl$_2$ transfer across the gas-

diffusion layer was not a limiting factor and should be higher than the gaseous product generation rate on the catalyst layer under the NO$_3^-$/Cl$^-$ concentration in common waste streams.

We then changed the mole ratio of NO$_3^-$ and Cl$^-$ ions concentration from 4:1 to 1:4 under the same NO$_3^-$ concentration (25 mM). The data in the right side of Fig. 4a show that the (NH$_4$)$_2$SO$_4$ generation rates and NH$_3$/Cl$_2$ recovery efficiencies remained relative stable between 556 ± 13 to 628 ± 19 g-(NH$_4$)$_2$SO$_4$·m$^{-2}$·d$^{-1}$ and 95 ± 1% to 98 ± 1% as the ratio of NO$_3^-$/Cl$^-$ decreased. Using the same high feed Cl$^-$ concentration (100 mM), a NaClO generation rate of 962 ± 46 g-NaClO·m$^{-2}$·d$^{-1}$ was obtained for the mole ratio of NO$_3^-$/Cl$^-$ of 1:4 and outperformed that (625 ± 40 g-NaClO·m$^{-2}$·d$^{-1}$) when NO$_3^-$/Cl$^-$ = 1:1. That could be attributed to the high feed NO$_3^-$ concentration leading to a high NH$_3$ yield, which in turn caused more un-separated NH$_3$ to react with the active chlorine in the waste stream[38], causing more Cl$_2$ loss. In conclusion, this membrane-free electrolyzer could consistently produce and recover NH$_3$/Cl$_2$ across various waste stream conditions.

Extended electrolysis experiments reveal that the electrocatalyst and gas diffusion layer were stable for over 100 h of synchronous production of (NH$_4$)$_2$SO$_4$ and NaClO with an average yield rate of 598 g-(NH$_4$)$_2$SO$_4$·m$^{-2}$·d$^{-1}$ and 182 g-NaClO·m$^{-2}$·d$^{-1}$, respectively (Fig. 4b). To enhance productivity, we expanded the area of individual electrode module from 9 cm$^2$ to 50 cm$^2$ and scaled up the reactor from a single module to a configuration of three tandem stacked modules (Fig. 4c), which constitutes a cumulative geometric electrode area of 300 cm$^2$. The cell potential for each module was consistently maintained at 3.0 V. The real reverse osmosis retentate from the Yuma Desalination

Plant in Arizona was further used as the feed waste stream, containing average concentrations of $NO_3^-$ and $Cl^-$ at ~11.8 mM and 54.1 mM (pH 6.6), respectively, alongside other co-existing contaminants such as $Ca^{2+}$, $Mg^{2+}$, $Na^+$, $SO_4^{2-}$ et al. A comprehensive analysis of the feed waste stream composition is provided in Supplementary Table 3. To boost the production of $(NH_4)_2SO_4$ and NaClO, 0.2-liter solutions of $H_2SO_4$ and NaOH were utilized as the trapping solutions for $NH_3$ and $Cl_2$, respectively. As shown in Fig. 4d, following a 6-h operation period, the stacked electrolyzer system yielded $83.8 \pm 16.7$ mM of $(NH_4)_2SO_4$ and $243.4 \pm 15.6$ mM of NaClO, accompanied by the nitrogen and chlorine average utilization efficiencies of 71% and 45%, respectively. The electrical consumption for the simultaneous production of $(NH_4)_2SO_4$ and NaClO was calculated at 7.1 kWh per aggregate kilogram of solid products. Concurrently, the residual concentrations of $NH_3/NH_4^+$, $Cl_2/$ $HClO/ClO^-$, and $NO_2^-$ in the treated reverse osmosis retentate were found at $0.32 \pm 0.19$ mM, $0.06 \pm 0.02$ mM, and $0.15 \pm 0.08$ mM (Fig. 4e), respectively. These values are below the relevant nitrogen- and chlorine-species regulatory limits for wastewater discharges into receiving water bodies. For instance, the World Health Organization (WHO) and the US Environmental Protection Agency (EPA) have not established a Maximum Contaminant Level (MCL) for ammonia, but common environmental limits for ammonia in surface water typically range between 0.02–2.32 mM. In addition, the MCL for free chlorine in drinking water, as stipulated by the US EPA, is set at 0.11 mM. The MCL for nitrite in drinking water is defined as 0.21 mM N by WHO[45] and 0.07 mM by the US EPA[46]. These findings underscore the viability of the stacked, membrane-free electrolyzer system for industrial-scale applications.

### Economic analysis and operation viability

A simple techno-economic analysis (TEA) was conducted to evaluate the profitability of this approach to synthesize ammonium sulfate and sodium hypochlorite using renewable energy sources (e.g., wind power, solar power, bioenergy, and hydroelectric) and the synthetic feed wastewater[47]. The TEA calculation was based on electricity cost of 5¢·kWh$^{-1}$ and the current market prices of ammonium sulfate ($533·ton$^{-1}$) and sodium hypochlorite ($958·ton$^{-1}$ for 60% purity)[48,49]. The computational contour plot depicted in Supplementary Fig. 11 clearly illustrates the impact of improvements in energy-related parameters (kg·$(NH_4)_2SO_4$/NaClO·kWh$^{-1}$), along with a reduction in the unit cost of electricity or a combination of both, in significantly mitigating the production costs of $(NH_4)_2SO_4$ and NaClO. The outcomes demonstrate the sensitivity of energy-related parameters to the $NO_3^-$/$Cl^-$ concentrations and ratios. Based on the laboratory-scale data and after deducting the associated electricity costs, the average profit attainable per metric ton for the obtained $(NH_4)_2SO_4$ and NaClO from synthetic waste streams with variable water chemistry parameters is projected to be $1550, as depicted in Supplementary Fig. 12. For the real RO stream, the projected profit is estimated at $2364 when scaled up to an industrial scale electrolyzer. This financial metric underscores the potential economic viability of this electrosynthesis process from the waste stream.

To produce different ammonium salts, we employed $HNO_3$, $H_3PO_4$, and a mixed acid solution comprising $HNO_3$, $H_2SO_4$, and $H_3PO_4$ in a 1:1:1 molar ratio for $NH_3$ capture. The results in Supplementary Fig. 13 reveal comparable $NH_3$ capture efficiencies across different acid solutions, underscoring the system's adaptability and flexibility in producing various ammonium salts. To avoid the use of hazardous chemicals, in situ acid/alkaline production via water electrolysis reaction was achieved with a proton exchange membrane-separated electrolyzer, featuring a stainless-steel mesh cathode and DSA anode, which generated the acid and alkaline solutions from a 0.1 M $Na_2SO_4$ feed. As shown in Supplementary Fig. 14, two types of operations were tested: two-stage (where acid/alkaline solutions are generated first and then used for $NH_3$/$Cl_2$ capture) and single-stage (where acid-base

solutions are generated simultaneously while capturing $NH_3/Cl_2$). Although $NH_3$ capture efficiency remained consistent, significant $Cl_2$ product loss (85%) was observed in the single-stage operation (Supplementary Fig. 15). This loss is likely because $NH_4^+$ cannot be readily oxidized at the anode, whereas $OCl^-$ can be reduced to $Cl^-$ at the cathode[16,38].

We further conducted experiments in a single-pass mode without recirculating the waste stream storage tank, detailed in Supplementary Figs. 16, 17. At flow rates of 5 mL·min$^{-1}$ and 25 mL·min$^{-1}$, the yields of $(NH_4)_2SO_4$ reached $64.51 \pm 3.21$ mM and $23.43 \pm 1.19$ mM, respectively, after 5 h, with $NH_3$ separation efficiencies of 54% and 71%. The discharged waste stream contained $NO_2^-$ and $NH_3$ concentrations ranging from $0.21 \pm 0.02$ mM to $1.07 \pm 0.06$ mM and $0.96 \pm 0.06$ mM to $5.48 \pm 0.18$ mM, respectively. The $Cl_2$ separation efficiencies exceeded 99%, maintaining residual active chlorine concentrations in the discharged waste stream below 0.01 mM throughout. To ensure compliance with regulatory limits, careful control of the flow rate is essential. These findings provide critical data for decision-makers and stakeholders to assess the economic benefits and potential applications of this membrane-free electrolyzer in chemical synthesis with waste streams.

## Discussion

This study showcases the simultaneous separation and recovery of $NH_3$ and $Cl_2$ from waste streams containing $NO_3^-$ and $Cl^-$ using a flow-type membrane-free electrolyzer. Within the electrolyzer, three primary stages are involved: (1) electrochemical conversion of $NO_3^-$ and $Cl^-$ ions into $NH_3$ and $Cl_2$; (2) vaporization of $NH_3$ and $Cl_2$ at the respective basic and acidic interfaces of the cathode and anode; and (3) interfacial extraction of $NH_3$ and $Cl_2$ at the electrode surface. The pairing of nitrate-reduction-to-ammonia with chloride-oxidation-to-chlorine evolutions eliminated undesired by-products, such as $H_2$ and $O_2$. The specially designed gas extraction electrode concurrently coupled electrosynthesis and product extraction, achieving the simultaneous generation and separation of $NH_3$ and $Cl_2$ on the same interface, thereby preventing significant product loss caused by redox reactions between $NH_3$ and $Cl_2$. Scale-up electrosynthesis using a stacked electrolyzer system with a geometric electrode area of up to 300 cm² and real reverse osmosis retentate waste stream was proven to be feasible. This work highlights the promise of combining $NH_3$ and $Cl_2$ production/separation using a straightforward electrolyzer configuration.

Future research should include the electrosynthesis and separation of a more diverse array of bulk and fine chemicals. In addition, future studies should explore integrating pre-concentration processes for low-concentration waste streams and optimizing reactor or catalyst layer designs, such as zero-gap electrolyzers, flow-through electrodes, or coupled porous adsorption materials to overcome mass transfer limitations. Electrosynthesis based on waste streams presents a cost-effective alternative to traditional waste removal processes, maximizing the value extracted from complex, abundant wastewater resources. Its on-site deployment at wastewater treatment facilities or pollution sources supports the circular economy, promotes energy sustainability, and enables zero liquid discharge. This approach offers substantial economic, environmental, and societal advantages.

## Methods
### Materials and reagents
Copper sulfate pentahydrate ($\geq 99$%), ruthenium dioxide nanoparticles (99.95%), sulfuric acid (98%), isopropanol (99.6%), sodium nitrate (98.8%), sodium chloride (99%), sodium sulfate (99%), sodium hydroxide (97%), nitric acid (69%–70%), hydrochloric acid (36.5%–38%), sulfamic acid (99%), p-aminobenzene sulphanilamide (98%), N-(1-Naphthyl) ethylenediamine dihydrochloride (96%), phosphoric acid (85%), salicylic acid ($\geq 99$%), sodium citrate dihydrate ($\geq 99$%), sodium

hypochlorite (5.65%-6%), and sodium nitroferricyanide (99%) were obtained from Thermo Fisher Scientific and used without further purification. The DPD-free chlorine reagent powder pillow was obtained from HACH Company. AvCarb GDS2230 substrate, Teflon PTFE DISP 30 Fluoropolymer Dispersion, Nafion 117 membrane, Vulcan XC 72 carbon black, and Nafion D-521 dispersion were purchased from Fuel Cell Store. Deionized (DI) water (18.2 MΩ cm) was applied throughout all experiments in this research.

### Fabrication of the Cu dendrite gas extraction electrode

A copper (Cu) dendrite electrocatalyst layer was deposited on a commercial carbon-based substrate (AvCarb GDS2230). The substrate consists of a carbon fiber layer (PTFE treated) and a carbon-based micro-porous layer (Supplementary Fig. 2). To enhance the anti-wetting properties of the gas extraction electrode, the substrate was further coated with another non-conductive PTFE hydrophobic layer on the PTFE treated carbon fiber layer side via air-brush spray using a 10 wt% PTFE solution and calcination operation (obtained by diluting Teflon PTFE DISP 30 Fluoropolymer Dispersion by DI water). The catalyst was deposited on the carbon-based micro-porous layer via an electrodeposition process in a typical three-electrode system. Briefly, a CHI 150 saturated calomel electrode (SCE), an $IrO_2 - RuO_2/Ti$ electrode (obtained from Yunxuan Metallic Materials Co. Ltd., China) (total size: 7 cm × 7 cm, area exposed to the electrolyte: 3 cm × 3 cm), and the substrate (total size: 4 cm × 4 cm, area exposed to the electrolyte: 3 cm × 3 cm) were used as reference electrode, counter electrode, and working electrode, respectively. The working electrode and counter electrode chambers were filled with 20 mL 0.1-M $CuSO_4 \cdot 5H_2O$ solution (prepared by pH = 2.0 ± 0.1 DI water, adjusted by 1 M $H_2SO_4$) and 20 mL 0.1-M $Na_2SO_4$ solution (pH = 7.0 ± 0.1), which were separated with a proton-exchange membrane (Nafion 117, total size: 7 cm × 7 cm, 183 μm in thickness, immersed in 0.1 M $Na_2SO_4$ solution overnight before use). All electrolytes were stored at room temperature, approximately 20 °C, and were utilized or disposed of within one week. Before catalyst deposition, the microporous layer side of each substrate was infiltrated with 200 μL isopropanol to improve the substrate's surface wettability. A constant potential (− 0.743 V vs SCE for 700 s) was applied to the working electrode by a CH Instruments 700E Potentiostat. After electrodeposition, the obtained Cu dendrite catalyst layer was rinsed with DI water and then dried in a 50 °C vacuum for 5 h. The catalyst loading was controlled to be 2.30 ± 0.05 mg·cm$^{-2}$, calculated by dividing the mass difference of the substrate before and after catalyst application by the electrode area exposed to the electrolyte.

### Fabrication of the RuO$_2$ anodic gas extraction electrode

The $RuO_2$ electrocatalyst layer was deposited on the same pretreated substrate via air-brush painting of the catalyst ink. The catalyst ink was prepared by mixing 10 mg of $RuO_2$ nanoparticles, 5 mg of Vulcan XC 72 carbon black, and 100 μL Nafion solution (D521 Nafion Dispersion at 5 wt%, containing ~ 4 mg Nafion) in 3.9 mL isopropanol. After sonication (50-60 Hz and 230 W) for 1 h, 2 mL of the catalyst ink was uniformly sprayed using an airbrush onto the substrate (total size: 4 cm × 4 cm)[50]. The $RuO_2$ catalyst layer-coated electrode was further air-dried overnight before testing. The catalyst loading was controlled to be 0.30 ± 0.03 mg·cm$^{-2}$, calculated by dividing the mass difference of the substrate before and after catalyst application by the total electrode size.

### Electrocatalyst-coated electrode characterization

The morphology and chemical composition of a prepared aqueous gas-extraction electrode was analyzed by JSM-7900F field emission scanning electron microscope (FE-SEM) (JEOL, Japan). The crystalline structures of the electrocatalysts were investigated by X-ray powder diffractometer (XRD) performed on a Philips, EMPYREAN, PANalytical Almelo with a Co Ka radiation (λ = 1.789 Å).

### Electrolyzer setup and operation

Supplementary Fig. 6 shows the major assembly procedure of the $NH_3$ trap channel, waste stream channel, and $Cl_2$ trap channel (all with length and width of 30 mm × 30 mm and depth of 10 mm in the flow cell, except for the waste stream channel with a depth of 20 mm) with the corresponding gas-extraction electrodes. Silicone gaskets (30 mm × 30 mm exposure window) ensured adequate sealing between each channel or end plate. All channels featured identical-sized inlets and outlets (4 mm OD; 2 mm ID) for electrolyte flow. The Cu-dendrite cathode and the $RuO_2$ anode separated the middle waste stream channel from the $NH_3$ trap channel and the $Cl_2$ trap channel, respectively. The electrocatalyst-coated side of the aqueous gas extraction electrodes faced the waste stream, whereas the PTFE gas diffusion layer side faced the $NH_3$ or $Cl_2$ trap channels. $NH_3$ and $Cl_2$ gases were extracted from the wastewater through the gas extraction electrodes' gas diffusion layer into the $NH_3$ or $Cl_2$ trap channels, respectively, due to the vapor pressure gradient of $NH_3$ and $Cl_2$ gases.

To evaluate synchronous electrosynthesis and separation of $NH_3$ and $Cl_2$ from waste stream, 50 ml synthetic wastewater solutions with different $NO_3^-$ and $Cl^-$ concentrations (pH = 7.0 ± 0.1, $Na_2SO_4$ = 0.1 M $NO_3^-$ = 10–100 mM, $Cl^-$ = 6.25–100 mM) were prepared and circulated between the waste stream channel and a feed tank at a flow rate of 25 mL·min$^{-1}$ using a peristaltic pump (MASTERFLEX L/S, Avantor, Radnor, US). Synthetic wastewater was prepared by adding target amounts of NaNO$_3$, NaCl, and Na$_2$SO$_4$ into 500 mL DI water to achieve the desired concentrations. The trap solutions were recirculated between the storage tank and the $NH_3$ trap channel with 50 mL pH = 1.0 ± 0.1 solution or the $Cl_2$ trap channel with 50 mL pH = 13.0 ± 0.1 solution at a flow rate of 25 mL·min$^{-1}$. Trap solutions of varying pH levels were prepared by adding 1 M $H_2SO_4$ or 1 M NaOH into 500 mL DI water and monitoring the pH value to achieve the desired acidity or alkalinity. All synthetic wastewater solutions and trap solutions were stored at room temperature, ~20 °C, and were utilized or disposed of within one week.

The individual $NH_3$ or $Cl_2$ electrosynthesis and separation were performed under a constant cell potential using a CH Instruments 700E Potentiostat at room temperature (~25 °C) and atmospheric pressure. These experiments were operated in a three-electrode configuration with Cu dendrite gas extraction electrode, $RuO_2$ anodic gas extraction electrode, and SCE serving as working electrode, counter electrode, and reference electrode, respectively. The potential measured was calibrated into a reversible hydrogen electrode (RHE) by:

$$E_{RHE} = E_{SCE} + 0.241V + 0.0591 \times pH \tag{5}$$

For synchronous electrosynthesis and separation of $NH_3$ and $Cl_2$ experiments, a DC power supply was used with total cell potentials ranging from 2.5 V to 3.5 V. The Cu dendrite gas extraction electrode and $RuO_2$ anodic gas extraction electrode served as cathode and anode, respectively. To record the current values of the electrolyzer operated under typical recycle mode and single pass mode, a CH Instruments 700E Potentiostat was utilized. These experiments were operated in a two-electrode system with the Cu dendrite gas extraction electrode and $RuO_2$ anodic gas extraction electrode serving as the working and counter electrodes, respectively. A constant cell potential of 3.0 V was maintained throughout these experiments. The collected current values are shown in Supplementary Figs. 7, 17.

The major products within the electrolyzer may have included nitrate-N, nitrite-N, ammonia-N, and free chlorine (hypochlorous acid and hypochlorite ion), which were analyzed using ultraviolet-visible (UV-vis) spectrophotometry as detailed in Supplementary Information, Section 10)[9]. To ascertain the ammonia source, control experiments were performed by adding or removing $NO_3^-$ to/from the synthetic wastewater solution. The findings confirmed that the produced ammonia originated exclusively from nitrate in the catholyte

rather than from ammonia-containing pollutants present in synthetic reagent raw materials, air, or human breath, used for the electrocatalytic layer. The relevant yield rate, separation efficiency, and energy consumption of obtained products were also calculated as shown in Supplementary Information, Section 11.

The real reverse osmosis retentate from the Yuma Desalination Plant in Arizona was also used as the feed waste stream to carry out the electrosynthesis experiment. The detailed composition analysis of the actual RO retentate was listed in Supplementary Table 3.

## Data availability

The data underlying the findings of this study are provided in the main text and Supplementary Information. Additional data related to the results discussed are available from the corresponding authors upon reasonable request. Source data are also provided as a Source Data file. Source data are provided in this paper.

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

## Acknowledgements

The study was financially supported by the NSF/BSF project (No. 2215387, received by W. Z.), New Jersey Water Resources Research Institute (No. G21AP10595-01, received by W. Z. and J. G.), and the 2024 NJIT's Technology Innovation Translation and Acceleration (TITA) Seed Grant program (received by W. Z.). The authors express their gratitude to Richard Bash and David M. Guerrero at the Yuma Desalination Plant in Arizona for their invaluable assistance with the shipping of RO retentate.

## Author contributions

J.G. conceived the project, designed/performed the experiment, and wrote the original draft. Q.M. conducted the SEM and XRD measurements. W.Z., Z.W., and B.R. were responsible for supervision, conceptualization, writing, reviewing, and editing.

## Competing interests

The authors declare no competing interests.
