## [Peer Review File · Nature Communications]

REVIEWER COMMENTS

Reviewer #1 (Remarks to the Author):

The authors describe a single chamber electrolyzer treating brine solution, integrated with electrolysis and separation of ammonia and chlorine by using dual gas diffusion electrodes. The electrochemical operation parameters were detailedly characterized and a stacked electrolyzer system were implemented to demonstrate stability. However, this study lacks scientific novelty for a high impact journal, please see below comments:

1. The concept of using gas diffusion electrodes to in-situ separate gas products such as ammonia have been well established in many previous papers (Environ. Sci. Technol. 2018, 52, 15, 8930–8938; Environ. Sci. Technol. 2021, 55, 11, 7674–7680; Energy Environ. Sci., 2021, 14, 1959–2008). It is not surprising to find good separation efficiency using typical gas diffusion electrodes for both ammonia and chlorine, especially with strong sulfuric acid and sodium hydroxide as trap solutions. There is no new separation mechanism or new electrode design for high separation efficiency, which did not qualify for a high impact journal.
2. The authors also need to consider the impact of side reactions of oxygen evolution and hydrogen evolution on overall system stability, faradic efficiency and cost-effectiveness. It's simply not possible to just have chlorine evolution and nitrate reduction reactions.
3. Also the system was tested at 3.0 V and proposed industrial current density exceeding 400 mA cm⁻², which remains questionable for practical applications. Gas diffusion electrodes would be easily wetted at high current density, especially with solutions on both sides. The paper would be more interesting if the authors could come up with strategy resolving this issue.
4. The use of expensive acid and base for just generating ammonium sulfate and sodium hypochlorite is not economically favorable. It would be great if the authors would try other trapping strategy such as negative pressure or other chemical free method to achieve a real sustainable pathway.
5. It should be noted that the effluent from the proposed system is not clean water, since there still exist complex ions such as phosphate, fluoride and potential toxic organic pollutants.

Reviewer #2 (Remarks to the Author):

General comments

The researchers developed a new type of flow electrolyzer that can synthesize and separate ammonia and chlorine from ultrafiltered concentrated water without using a membrane. The electrolyzer contains carbon-polytetrafluoroethylene sets with immobilized copper and ruthenium oxide particles. Ammonia and chlorine are generated using the NO₃RR and CER reactions and collected using corresponding trap solutions. The system was able to produce high concentrations of (NH₄)₂SO₄ (83.8 mM) and NaClO (243.4 mM). The cost of electricity per kilogram of solid product was about 7.1 kWh, and the pollutants in the effluent were within acceptable limits.

This study presents a new method for regenerating energy from wastewater and collecting and reusing

industrial products such as chlorine and ammonia. It shows promising applications in reverse osmosis concentrated water. However, the article may benefit from a more detailed exploration of mechanisms and innovative aspects. Some of the conclusions rely on simple chlorine/ammonia yield rates, which could impact the overall credibility and innovation of the work. Therefore, I strongly recommend that the authors revisit and refine their research by addressing these concerns. The current version of the paper is not recommended for publication in Nature Communications. I would be happy to re-evaluate the updated manuscript after the authors have thoughtfully addressed these issues.

Specific comments

1. Could you please explain the relationship between the volcano-like response of NH₃ separation efficiency and cathode potential in Figure 1e? Also, does the H₂ evolution side reaction significantly affect the separation of NH₃? What is the H₂ coverage during this process? And is there any co-overflow of H₂ and NH₃ occurring in this process?
2. The author suggests in the manuscript that a rapid waste flow channel can prevent contact between NH₃ and Cl₂ in wastewater, inhibiting chlorination reactions leading to N₂ formation. Although this method can reduce the reactions between the two and enhance the collection efficiency of NH₃ and Cl₂, it cannot eliminate the losses associated with N₂ and chlorination. Could the author quantify these losses through mass balance calculations and demonstrate whether such losses are within acceptable limits?
3. Does the concentration of the trap solution significantly affect the capture of NH₃? Besides sulfuric acid, how do other trap solutions perform in capturing NH₃? It is suggested to include relevant data analysis of trap solutions for NH₃ capture to ensure the scalability of NH₃ recovery in the system.
4. In line 192, the author mentions the main reactions occurring in the electrolytic cell, including the reduction of HClO. The preceding section has demonstrated the transformation of hypochlorite into NaClO₄ in the trap solution. Please confirm the chemical forms involved in this part of the study. Additionally, in line 203, the author states, "N₂ was dominant because the reaction between NH₃/NH₄⁺ and HClO/ClO⁻ generates N₂ (4.2*10⁶ M⁻¹·s⁻¹) faster than NO₃⁻ (0.1–0.7 M⁻¹·s⁻¹)." During this process, HClO cannot react with NO₃⁻, so it is recommended to revise the narrative in this section.
5. In Figure 3c, the author posits that active chlorine oxidizes NH₃ to N₂, thereby influencing the conversion of NO₃⁻ and NO₂⁻ to NH₃. This conclusion is contentious because, during the process, the concentration of NO₂⁻ rapidly decreases, transforming into a large quantity of NO₃⁻ and a small amount of NH₃. Moreover, the rate of increase in NO₃⁻ concentration is significantly higher than the chlorination rate of NH₃. Therefore, it is suggested that the primary reason influencing the nitrogen forms is likely the oxidation of NO₂⁻ to NO₃⁻. It is recommended that the author compare the reaction kinetics parameters for the conversion of different nitrogen forms to elucidate the dominant reactions in the complex system.
6. In lines 255-257, the author suggests that the higher concentrations of NO₃⁻ and Cl⁻ result in increased NH₃/Cl₂ yields, indicating that mass transfer on the gas diffusion layer is not a limiting factor. However, this assertion needs further confirmation, as the author only explores the concentration increase from 10 mM to 100 mM for NO₃⁻ and Cl⁻. This range may not adequately reveal the limitations imposed by mass transfer on the increase in yields. Alternatively, the author may consider citing references that provide evidence for the typical concentrations of NO₃⁻ and Cl⁻ in concentrated water within this specified range.
7. It is recommended that the author conduct a cost assessment of the practical reverse osmosis retentate recovery system. This evaluation will help demonstrate whether the obtained values of

(NH₄)₂SO₄ and NaClO surpass the total costs of trap solution and energy consumption, thereby emphasizing the relevant advantages of resource recovery in this system.

8. The author's exploration of the mechanism appears to be insufficiently in-depth, focusing more on quantifying the yields and recovery efficiencies of NH₃ and Cl₂. Additionally, the study lacks an extended investigation into the practical applications of the recovered NH₃ and Cl₂ in this system. While we acknowledge the engineering application value of this work, the research content seems somewhat one-sided. To enhance the study, it is suggested that the author consider incorporating economic feasibility calculations or proposing improved methods to prevent reactions between NH₃ and Cl₂, thereby further increasing the overall yield. This expansion would contribute to elevating the significance of the work.

9. The author should include information about the pH variations during the system reaction process, as side reactions can significantly impact the electrolyte pH. These variations may lead to changes in the trapping solution's performance in capturing gases. Additionally, the separation of NH₃ and Cl₂ recovered from concentrated water and the trap solution should be considered in subsequent investigations to ensure the regeneration and sustainable use of the system.

10. The schematic diagram for the TOC requires thorough revision. The internal reaction processes within the green background are not clearly displayed, and the diagram does not effectively serve its purpose of providing an overview and introduction to the content of the article. ly serve its purpose of providing an overview and introduction to the content of the article.

Reviewer #3 (Remarks to the Author):

The paper, 'Direct electrosynthesis and separation of ammonia and chlorine from waste streams via a stacked membrane-free electrolyzer' describes an interesting waste-water electrolysis setup with nitrate reduction occurring on the cathode and chlorine evolution on the anode. The authors measure ammonia yield/separation and chlorine yield/separation, as well as side products and reactions. Further, the system was then scaled up to a 300cm² stack. The system is thoroughly investigated for different inlet concentrations of NO₃⁻ and Cl⁻ and electrode voltage over a number of hours and the conclusions are supported by the results.

However, upon reviewing the manuscript there are a few confusingly written experimental descriptions and questions/topics raised:

Major comments

1) There are many illustrations and schematics in the paper, such as Fig1a, Fig1b Fig2a, Fig4c, but I did not find these useful in understanding the electrolyser setup as Fig1a did not have enough information for the reader, Fig1b is awkwardly split into two, Fig2a mentions a 'cathodic membrane' and 'anodic membrane' when this is a membrane-less device. The most useful figure, in my opinion, is Figure S4 and this should be in the main paper, provided more detail is added to this too. For instance, 'cathodic aqueous gas extraction electrode' is too vague, it would be helpful to have the substrate, catalyst etc given on here. This looks like three layers but I can only identify two from the experimental section, being the catalyst and PTFE, what is the third layer?

2) There needs to be substantially more information given in the paper about how the gas-liquid

separation substrates work and their materials/properties, is this a simple hydrophobic PTFE substrate or is it an AvCarb GDL? How is electronic conductivity ensured through the PTFE to the endplate?

3) There is inconsistent language which is confusing for the reader and terms are not properly defined, for instance, 'ammonia trap channel' vs 'ammonia recovery channel' seem to be used interchangeably.

4) If separation of gas-liquid occurs in the catalyst layer, how is the pH maintained by adding H₂SO₄ and NaOH to the trap channel as described in the materials section?

5) No polarisation curves are provided in the manuscript for readers to understand the voltage/current relationship, some current densities are provided in the supplementary (Figure S1) but this is not sufficient

6) What are your faradaic efficiencies?

Minor comments

1) There are some typos in the manuscript, please read and edit carefully.

2) Please also describe the 'recovery channel', is this simply a flow channel or is there a specific property to it?

3) If H₂SO₄ and NaOH are added to the electrolyser, how much of this exits in the 'clean' water and does this still meet the wastewater regulatory limits?

Reviewer #4 (Remarks to the Author):

I very much enjoyed reading this manuscript since it is not the typical performance based catalyst development. This said, there are several things that would have to be resolved and thus I cannot support publication of this study.

1) One of the biggest issue with the manuscript is that the authors show no composition analysis of the actual waste water that is used herein. This might lead to reproducibility issues since additional ions can hamper the overall process. Along this line, it is questionable how stable the performance is against other anions and how they affect the overall performance. This needs to be done however to show the robustness of the process. In addition, it is not a good sign when only in the experimental part it is hidden that the authors utilised synthetic waste water solutions. The experiments need to be performed with real life ones to show the robustness of the system. Otherwise the story is simply made up and it has no relation to any waste water treatment system.

2) In their introduction, the authors claim that industrial wastewater has a high salinity. This is too special and is not true for all wastewater. A more specialised phrasing is necessary.

3) While multiple graphics/figures show error bars, these are needed for all experiments and should be displayed accordingly.

4) The authors mainly rely on their explanations on the analytics reported after 5 hours. This is not enough time to make a reasonable statement. As it looks like from the data, the system did not yet reach a steady state performance and thus no statement on performance should be provided. Likewise, this

referee wants to know how the system behaves in single pass mode when no recirculation of the storage tanks is used. This is the more likely scenario and it is important to show the performance in a continuous flow.

5) The authors claim that after their process they obtain clean water. I do not see any evidence for this since no detailed analysis to underline this statement is presented.

6) I am missing typical U/I and U or I/t curves for all provided experiments. What is the typical current density the authors measure at? This is important to judge the energy efficiency of the system and ohmic losses.

7) Only later on it seems that the authors used water from a desalination plant. However, also here, the composition is not specified and would require a thorough analytics to show its exact composition.

8) The calculated electrical consumption of the process should be taken with care. There are no downstream processes reported herein to remove solvents etc. which can be a central part of the energy consumption. This needs to be included as an analysis.

9) Figure 4b: Why does the data shown look so wavy. It almost looks like data is added in periods? What happens when the individual rate goes down during the time of operation?

10) In the last sentence of the conclusion the authors finally mention a current density. This is, however, misleading since no other current density is reported throughout the manuscript and it is unclear where such a number is obtained from and how it compares with the reported data.

Response to Reviewer #1

The authors describe a single chamber electrolyzer treating brine solution, integrated with electrolysis and separation of ammonia and chlorine by using dual gas diffusion electrodes. The electrochemical operation parameters were detailly characterized and a stacked electrolyzer system were implemented to demonstrate stability. However, this study lacks scientific novelty for a high impact journal, please see below comments.

Reply: Thank you for your valuable feedback on our manuscript. We understand your concerns regarding the perceived lack of scientific novelty in our study for a high-impact journal. In response, we have undertaken a thorough revision to more effectively highlight the innovative aspects of our research and its implications for the field. How we accomplished this is detailed in the responses that follow.

1. The concept of using gas diffusion electrodes to in-situ separate gas products such as ammonia have been well established in many previous papers (Environ. Sci. Technol. 2018, 52, 15, 8930–8938; Environ. Sci. Technol. 2021, 55, 11, 7674–7680; Energy Environ. Sci., 2021, 14, 1959–2008). It is not surprising to find good separation efficiency using typical gas diffusion electrodes for both ammonia and chlorine, especially with strong sulfuric acid and sodium hydroxide as trap solutions. There is no new separation mechanism or new electrode design for high separation efficiency, which did not qualify for a high impact journal.

Reply: We acknowledge established precedent for utilizing gas diffusion electrodes in NH₃ extraction, as highlighted in prior studies (Environ. Sci. Technol. 2018, 52, 15, 8930–8938; Environ. Sci. Technol. 2021, 55, 11, 7674–7680) and CO₂ delivery (Energy Environ. Sci., 2021, 14, 1959–2008). However, our work diverges significantly in the following approaches and objectives:

(1) **Significance of Integrated Electrosynthesis and Separation:** Different from delivery of gaseous reactants (such as CO₂ and N₂) via a gas diffusion layer, as described in the literature, our study emphasizes the practicality and economic viability of electrosynthesis and separation of ammonia and chlorine within a single electrode module. This dual process is distinct from

reported "electrochemical stripping", which separates *existing NH₃ in waste streams* by generating alkaline cathodic interface to achieve the conversion of NH₄⁺ ion to NH₃ gas. No electrosynthesis process and value-added product conversion are involved in the literature. In contrast, the nitrate-to-ammonia reaction in our process, ($\text{NO}_3^- + 6\text{H}_2\text{O} + 8\text{e}^- \rightarrow \text{NH}_3 + 9\text{OH}^-$) offers two *novel, synergistic effects: (a) transferring nitrate, a pollutant, to value-added ammonia and (b) transferring the proton consumption during this reaction into the driving force for the phase transfer of NH₄⁺/NH₃ group.*

(2) **Novelty in Cl₂ Electrosynthesis and In-situ Separation:** To our knowledge, this is *the first study of Cl₂ electrosynthesis coupled directly with in-situ separation.* This novel aspect of our research underscores the unexplored potential in this area. This unique electrolyzer design pairs cathodic and anodic reactions without using ion-selective or -exchange membranes; the approach can potentially be used to synthesize various gaseous products, including volatile fatty acids (VFAs) and in water-splitting electrolysis to mitigate the risks associated with H₂ and O₂ mixing. In conclusion, our work goes well-beyond the established use of gas diffusion electrodes for separation. It represents a novel strategy that leverages electrosynthesis from waste streams for efficient and sustainable chemical production.

2. The authors also need to consider the impact of side reactions of oxygen evolution and hydrogen evolution on overall system stability, faradic efficiency and cost-effectiveness. It's simply not possible to just have chlorine evolution and nitrate reduction reactions.

Reply: We added new data on Faradaic Efficiency and current density in Supplementary **Table 2**; the new data provide a comprehensive view of these factors. **Fig. 4b** shows the production rates of the two products to assess system stability. Additionally, analysis of cost-effectiveness is included in **Figures S11** and **S12**. Despite some hydrogen and oxygen evolution side reactions, the system still exhibited good stability and cost-effectiveness. Certainly, future research may employ more-efficient catalysts and strategies to avoid any side reactions and, thus, enhance the system's cost-effectiveness.

In Line 340-344, page 22, we added the following:

Based on the laboratory-scale data and after deducting the associated electricity costs, the average profit attainable per metric ton for the obtained $(\text{NH}_4)_2\text{SO}_4$ and NaClO from synthetic waste streams with variable water chemistry parameters is projected to be \$1550, as depicted in Supplementary Fig. 12. For the real RO retentate waste stream, the projected profit is estimated at \$2364 when scaled up to an industrial scale electrolyzer.

Table S2. Summary of current density and NH_3/Cl_2 FE of different NO_3^- and Cl^- concentrations and ratios under the fixed 3.0 V cell potential.

Concentrations and ratios	Current densities	FE for NH_3	FE for Cl_2
$\text{NO}_3^-:\text{Cl}^-$ concentration ratio=1			
10 mM NO_3^- , 10 mM Cl^-	4.00±0.22 mA·cm ⁻²	69.9±1.8%	2.7±0.6%
25 mM NO_3^- , 25 mM Cl^-	5.67±0.56 mA·cm ⁻²	79.1±1.1%	9.6±0.2%
50 mM NO_3^- , 50 mM Cl^-	8.44±0.67 mA·cm ⁻²	87.8±2.0%	25.6±0.9%
100 mM NO_3^- , 100 mM Cl^-	13.56±1.22 mA·cm ⁻²	92.6±0.5%	36.9±0.1%
$\text{NO}_3^-:\text{Cl}^-$ concentration ratios from 4:1 to 1:4			
25 mM NO_3^- , 6.75 mM Cl^-	5.67±0.44 mA·cm ⁻²	74.3±0.7%	0.5±0.1%
25 mM NO_3^- , 12.5 mM Cl^-	5.78±0.67 mA·cm ⁻²	75.4±0.4%	0.7±0.0%
25 mM NO_3^- , 25 mM Cl^-	5.67±0.56 mA·cm ⁻²	79.1±1.1%	9.6±0.2%
25 mM NO_3^- , 50 mM Cl^-	6.78±0.33 mA·cm ⁻²	63.7±2.8%	18.1±0.5%
25 mM NO_3^- , 100 mM Cl^-	8.00±0.89 mA·cm ⁻²	54.7±2.5%	36.1±0.7%

Fig. 4b. Performance of the membrane-free electrolyzer without changing the cathodic/anodic modules at a cell potential of 3.0 V, 25 mM NO_3^- , 25 mM Cl^- . Each operation cycle last 5 h

with >85 NO_3^- converted to NH_3 and $>90\%$ NH_3 separated from the electrolyte.

Figure S11. Production cost of $(\text{NH}_4)_2\text{SO}_4$ (a) and NaClO (b) as a function of energy-related parameters and unit electricity cost.

Figure S12. The market prices of the obtained products under \$1000 electricity input with a set base electricity cost of 5 ¢/kWh. The calculation is based on the recovered $(\text{NH}_4)_2\text{SO}_4$ and NaClO in the NH_3/Cl_2 trap channels.

3. Also the system was tested at 3.0 V and proposed industrial current density exceeding 400 mA cm^{-2} , which remains questionable for practical applications. Gas diffusion electrodes would be easily wetted at high current density, especially with solutions on both sides. The paper would be more interesting if the authors could come up with strategy resolving this issue.

Reply: We appreciate the reviewer's insights regarding the operational parameters and potential practical application concerns. We analyze their impacts in two ways.

(1) **Cell Voltage and Industrial Current Density:** The choice of a 3.0-V cell voltage was made to align with typical NO_3^- and Cl^- concentrations in waste streams, as detailed in Lines 248-251, Page 16 ,and **Table S3**. Our objective was to balance the yield rate and energy consumption for practical NH_3 and Cl_2 electrosynthesis from waste streams, given that the prevalent ionic strength supports a relatively low cell voltage. To operate the electrolyzer using higher current density, additional preconcentration steps would be required to enhance NO_3^- and Cl^- concentrations. Based on the enhanced ionic strength, a higher cell voltage will be applied. Our further research will direct towards this challenge. We revised the conclusion to more accurately reflect the limitations and applicability concerning the current densities observed in our study.

Line 268-271, page 18.

We first increased the same molar concentrations of NO_3^- and Cl^- in the feed solution from 10 mM (a typical level for most industrial wastewater⁴⁴) to 100 mM found in brine wastewater from ion exchange or reverse osmosis processes,²¹ using a fixed cell potential of 3.0 V.

Line 385-388, page 24

Future research should include the electrosynthesis and separation of a more diverse array of bulk and fine chemicals. Additionally, it should investigate the integration of preconcentration processes for commonly encountered low-concentration waste streams to overcome mass transfer limitations.

Table S3. The characteristics of the obtained reverse osmosis (RO) retentate obtained from a RO plant for surface and ground water treatment.

Component	Concentration (mg L^{-1})
Conductivity	9790 uS cm^{-1}
pH	6.57 (measured at 19.5°C)
Nitrate as N	164.7 (~11.8 mM)
Aluminum	0.07
Barium	0.04
Bicarbonate as CaCO_3	11.3
Boron	1.47
Calcium	293

Calcium as CaCO ₃	775
Chloride	1920 (~54.1 mM)
Chromium	0.15
Fluoride	<20
Iron	1.53
Magnesium as CaCO ₃	650
PO ₄ ⁻ as P	<20
Potassium	32.1
Silicon Dioxide	26.0
Sodium	1840
Strontium	4.24
Sulfate	2680

(2) **Mitigating Electrolyte Flooding at High Current Densities:** Regarding the concern of electrolyte flooding at high current densities, particularly with solutions present on both sides of gas diffusion electrodes, our prior research has laid the groundwork. For example, we proved that a design using a relatively hydrophobic electrocatalyst layer could accelerate the electrosynthesis and gaseous products separation rates, enhance the long-term performance stability, and prevent flooding as shown in **Figure 1** and **2** below. Besides, another non-conductive PTFE hydrophobic layer coating on the back of the electrode module was employed in this research (**Figure S2** in Supporting Information). These operations can sustain a strong three-phase boundary and ensure the long-term stability of the electrode module, effectively preventing flooding during operation. Thus, the same electrode module interface design could be used to prevent electrolyte flooding or wetting issues.

Figure 1 for response letter. (a, b) Illustration of the two different electrocatalyst layers with or without the presence of the blended PTFE nanoparticles in the CuO nanoparticle layer. (c, d) Cross-sectional SEM images of PTFE-CuO/C and CuO/C electrodes. (e) Stability test of the CuO/C and PTFE-CuO/C electrocatalysts at a constant current density of 100 mA cm⁻². The electrolyte was replaced by a new solution of 100 × 10⁻³ M NaNO₃ + 0.5 M Na₂SO₄ for each cycle.¹

Figure 2 for response letter. g, The demonstration diagram of the hydrophobic dendrite electrocatalyst layer interface that partial contact with the electrolyte, avoiding the flooding problem. **h**, Cross section view of hydrophobic gas extraction electrode when immersed in electrolyte.²

Figure S2. The illustration of the pretreatment of substrate, catalyst coating, and connection between conductive sheet and obtained electrode.

References cited above:

1. Gao, Jianan, et al. "Decoupling Electron - and Phase - Transfer Processes to Enhance Electrochemical Nitrate-to-Ammonia Conversion by Blending Hydrophobic PTFE Nanoparticles within the Electrocatalyst Layer." *Advanced Energy Materials* 13.9 (2023): 2203891.
2. Gao, Jianan, et al. "Coupling Curvature and Hydrophobicity: A Counterintuitive Strategy for Efficient Electroreduction of Nitrate into Ammonia." *ACS nano* 2024, 18, 14, 10302–10311.

4. The use of expensive acid and base for just generating ammonium sulfate and sodium hypochlorite is not economically favorable. It would be great if the authors would try other trapping strategy such as negative pressure or other chemical free method to achieve a real sustainable pathway.

Reply: (1) Alternative Trapping Strategy: We recognize the potential of employing a negative pressure mechanism on the reverse side of the electrode module as a promising strategy. This approach also may relieve electrolyte flooding concerns and lead to purer NH_3 and Cl_2 products. Actually, we tried the negative pressure trapping module for NH_3/Cl_2 separation, as shown in **Figure 3** below. Using a weak negative pressure, the separation efficiency of NH_3/Cl_2 was suboptimal. Conversely, a strong negative pressure achieved similar separation efficiencies as those under filled liquid conditions but posed significant challenges to the long-term stability of the electrode assembly. In certain applications, like producing liquid fertilizers and disinfectants, trapping and converting the separated NH_3 and Cl_2 into target liquids using specific solvents proves to be more convenient.

Figure 1. for response letter. Picture of negative pressure trapping module.

(2) Alternative for Acid/Base Generation: We tried an additional method to circumvent the utilization of expensive acids and bases for NH_3/Cl_2 trapping. We experimented with an electrochemical approach to generate acid (from the anodic channel) and base (from the

cathodic channel) solutions by introducing a 0.1 M Na₂SO₄ solution for effective NH₃ and Cl₂ trapping.

(3) Versatility and Scalability of Liquid Products Production. To further validate the versatility and scalability of NH₃ recovery in our system, but also increase the variety of obtained liquid nitrogen fertilizers, we further explored NH₃ capture using alternative acidic solutions, such as HNO₃, H₃PO₄, and a mixed acid solution comprising H₂SO₄, HNO₃, and H₃PO₄ in a 1:1:1 molar ratio with the same initial pH of 1. These additional data and the comparative analyses of various trap solutions for NH₃ capture are documented in the "Economic Analysis" section and illustrated in Supplementary **Figures 13-15**.

Line 347-359, page 22.

To produce different ammonium salts, we employed HNO₃, H₃PO₄, and a mixed-acid solution comprising HNO₃, H₂SO₄, and H₃PO₄ in a 1:1:1 molar ratio for NH₃ capture. The results in Supplementary Fig. S13 reveal comparable NH₃-capture efficiencies across different acid solutions, underscoring the system's adaptability and flexibility in producing various ammonium salts. To avoid the use of hazardous chemicals, in-situ acid/alkaline production via water electrolysis reaction was achieved with a proton exchange membrane-separated electrolyzer, featuring a stainless-steel mesh cathode and DSA anode, which generated the acid and alkaline solutions from a 0.1 M Na₂SO₄ feed. As shown in Supplementary Fig. S14, two types of operations were tested: two-stage (where acid/alkaline solutions are generated first and then used for NH₃/Cl₂ capture) and single-stage (where acid-base solutions are generated simultaneously while capturing NH₃/Cl₂). Although NH₃ capture efficiency remained consistent, significant Cl₂ product loss (85%) was observed in the single-stage operation (Supplementary Fig. 15). This loss is likely because NH₄⁺ cannot be readily oxidized at the anode, whereas OCl⁻ can be reduced to Cl⁻ at the cathode.^{16, 38}

Figure S13. The comparison of various acid solution for NH₃ capture. Specific testing conditions: electrolyte channel: 25 mM NaNO₃, 25 mM NaCl, 0.1 M Na₂SO₄, pH 7.0; cell voltage 3.0 V. The flow rates of all electrolytes are 25 mL·min⁻¹.

Figure S14. (a) two-stage operation, where the acid/base was generated in a proton exchange membrane (PEM)-separated electrolyzer and then pumped into another electrochemical reactor for NH₃/Cl₂ separation (batch mode). (b) one-stage operation, where the acid/base was generated and continuously pumped into the electrochemical reactor for NH₃/Cl₂ separation (continuous mode).

Figure S15. The comparison of two-stage operation and one-stage operation for NH₃ (a) and Cl₂ (b) capture. Specific testing conditions: electrolyte channel: 25 mM NaNO₃, 25 mM NaCl, 0.1 M Na₂SO₄, pH 7.0; cell voltage 3.0 V. The flow rates of all electrolytes are 25 mL·min⁻¹.

5. It should be noted that the effluent from the proposed system is not clean water, since there still exist complex ions such as phosphate, fluoride and potential toxic organic pollutants.

Reply: Yes, the effluent from our proposed system cannot be considered “clean water” due to the presence of other contaminations in the feeding waste stream as only the concentrations of NO₃⁻ and Cl⁻ in the feed waste stream were reduced, Certain other pollutants will meet discharge standards: namely NH₃/NH₄⁺ (0.3 mM), NO₂⁻ (0.2 mM), and Cl₂/HClO/ClO⁻ (0.1 mM), which will comply with discharge regulations for nitrogen- and chlorine-containing species from the discharged real reverse osmosis retentate waste stream. We revised the wording in the “ABSTRACT” and “Large-scale electrosynthesis using a stacked electrolyzer system” sections to clarify this point and ensure there is no overstatement of our findings. Additionally, the annotations in **Fig. 2a** have been updated to reflect this clarification.

Line 22-25, page 2	
Original	Revised
This yielded high concentrations of (NH ₄) ₂ SO ₄ (83.8 mM) and NaClO (243.4 mM) at an electrical cost of 7.1 kWh per	This yielded high concentrations of (NH ₄) ₂ SO ₄ (83.8 mM) and NaClO (243.4 mM) at an electrical cost of 7.1 kWh per

kilogram of solid products, while maintaining minimal residual pollutants ($\text{NH}_3/\text{NH}_4^+$: 0.3 mM, NO_2^-: 0.2 mM, $\text{Cl}_2/\text{HClO}/\text{ClO}^-$: 0.1 mM) in the waste stream and meeting wastewater discharge regulations.	kilogram of solid products, while residual $\text{NH}_3/\text{NH}_4^+$ (0.3 mM), NO_2^- (0.2 mM), and $\text{Cl}_2/\text{HClO}/\text{ClO}^-$ (0.1 mM) in the waste stream meet the wastewater discharge regulations for nitrogen- and chlorine-species.
Line 305-309, page 20	
Original	Revised
Concurrently, the residual concentrations of $\text{NH}_3/\text{NH}_4^+$, $\text{Cl}_2/\text{HClO}/\text{ClO}^-$, and NO_2^- in the treated reverse osmosis retentate were found at 0.32 ± 0.19 mM, 0.06 ± 0.02 mM, and 0.15 ± 0.08 mM (Fig. 4e), respectively. These values are below the regulatory limits for wastewater discharges into receiving water bodies.	Concurrently, the residual concentrations of $\text{NH}_3/\text{NH}_4^+$, $\text{Cl}_2/\text{HClO}/\text{ClO}^-$, and NO_2^- in the treated reverse osmosis retentate were found at 0.32 ± 0.19 mM, 0.06 ± 0.02 mM, and 0.15 ± 0.08 mM (Fig. 4e), respectively. These values are below the relevant nitrogen- and chlorine-species regulatory limits for wastewater discharges into receiving water bodies.

Fig. 2 | The performance of synchronous electrosynthesis and separation of NH_3 and Cl_2 of the flow-type membrane-free electrolyzer. a, Schematics and configuration of this flow-type membrane-free electrolyzer for electrochemical synthesis and *in-situ* recovery of ammonium sulfate and hypochlorous acid from waste streams.

Response to Reviewer #2

The researchers developed a new type of flow electrolyzer that can synthesize and separate ammonia and chlorine from ultrafiltered concentrated water without using a membrane. The electrolyzer contains carbon-polytetrafluoroethylene sets with immobilized copper and ruthenium oxide particles. Ammonia and chlorine are generated using the NO_3RR and CER reactions and collected using corresponding trap solutions. The system was able to produce high concentrations of $(\text{NH}_4)_2\text{SO}_4$ (83.8 mM) and NaClO (243.4 mM). The cost of electricity per kilogram of solid product was about 7.1 kWh, and the pollutants in the effluent were within acceptable limits.

This study presents a new method for regenerating energy from wastewater and collecting and reusing industrial products such as chlorine and ammonia. It shows promising applications in reverse osmosis concentrated water. However, the article may benefit from a more detailed exploration of mechanisms and innovative aspects. Some of the conclusions rely on simple chlorine/ammonia yield rates, which could impact the overall credibility and innovation of the work. Therefore, I strongly recommend that the authors revisit and refine their research by addressing these concerns. The current version of the paper is not recommended for publication in Nature Communications. I would be happy to re-evaluate the updated manuscript after the authors have thoughtfully addressed these issues.

Reply: Thank you for your constructive feedback on our manuscript. We appreciate the opportunity to refine our work based on your insights and are committed to addressing the concerns raised to enhance the quality and clarity of our research.

1. Could you please explain the relationship between the volcano-like response of NH_3 separation efficiency and cathode potential in Figure 1e? Also, does the H_2 evolution side reaction significantly affect the separation of NH_3 ? What is the H_2 coverage during this process? And is there any co-overflow of H_2 and NH_3 occurring in this process?

Reply: The observed volcano-like response in NH_3 separation shows the interplay of the NH_3 -production and hydrogen-evolution reaction (HER) as competing side reaction. At certain cathodic potentials, the efficiency of NH_3 separation is maximized due to optimal

electrochemical conditions that favor NH_3 evolution over H_2 . However, beyond these potentials, the increased propensity for H_2 evolution begins to interfere with NH_3 separation and caused the "volcano" shape of the NH_3 separation efficiency. **Figure 4** shows the presence of bubbles on the back side of the electrode module. The transferred NH_3 gas is converted into NH_4^+ ions in the acid solution. Thus, the stable bubbles are attributed to H_2 . Therefore, it's crucial to control the cathodic potential to either avoid excessive H_2 generation or consider collecting H_2 as an additional value-added product, which will be carried out in our further research to enhance the overall sustainability and efficiency of the process.

Figure 4. Visible bubble extraction at the gas diffusion layer side of the electrode module.

We included new data on the Faradaic efficiency of various products under different cathodic potentials to quantitatively assess the dynamic evolution and co-overflow ratio of H_2 and NH_3 . The discussion on the volcano-like response of NH_3 separation efficiency in relation to cathode potential is further elaborated.

Line 123-129, page 8.

As depicted in **Fig. 1b**, increasing the cathodic potential from -0.50 to -0.80 V vs. RHE corresponded to a surge in the NH_3 yield rate, from 9.8 ± 1.1 to $49.4 \pm 0.7 \times 10^{-10}$ $\text{M-NH}_3 \cdot \text{cm}^{-2} \cdot \text{s}^{-1}$, accompanied by an upswing in the NH_3 transfer rate. Beyond -0.80 V vs. RHE, the NH_3 yields

stabilized around $50 \times 10^{-10} \text{ M-NH}_3 \cdot \text{cm}^{-2} \cdot \text{s}^{-1}$, but its transfer rate kept diminishing. This phenomenon can be ascribed to the enhanced Faradaic Efficiency (FE) for H_2 production at elevated cathodic potentials (Supplementary Fig. S5). The concurrent efflux of H_2 competes for the gas transfer channels with NH_3 , resulting in a diminished NH_3 transfer rate.

Figure S5. The Faradaic efficiency of various products under different cathodic potentials.

2. The author suggests in the manuscript that a rapid waste flow channel can prevent contact between NH_3 and Cl_2 in wastewater, inhibiting chlorination reactions leading to N_2 formation. Although this method can reduce the reactions between the two and enhance the collection efficiency of NH_3 and Cl_2 , it cannot eliminate the losses associated with N_2 and chlorination. Could the author quantify these losses through mass balance calculations and demonstrate whether such losses are within acceptable limits?

Reply: The relevant data for products loss is shown in Figure 2e (the green points and line). The relevant discussion is provided below for the reviewer's convenience.

Line 179-186, page 12.

When the cell potential increased from 2.5 V to 3.5 V, the average recovery efficiency (pink data points) for NH_3 and Cl_2 improved from 90% to 96% as Fig. 2e and Supplementary Fig. 8

indicate. The total loss of the produced NH_3 and Cl_2 (green data points) varied between 12% to 5%, with the minimum value located at 3.0 V. These results validate the membrane-free electrolyzer's effectiveness in co-electrosynthesizing NH_3 and Cl_2 with high efficiency and acceptable product loss. Furthermore, the interaction between residual nitrogen and chloride species in the waste stream resulted in the formation of N_2 and Cl^- as final products, further minimizing the residual harmful products such as $\text{NH}_3/\text{NH}_4^+$ and $\text{Cl}_2/\text{HClO}/\text{ClO}^-$.

Fig. 2 | The performance of synchronous electrosynthesis and separation of NH_3 and Cl_2 of the flow-type membrane-free electrolyzer. e, The average recovery efficiencies and the sum of the product loss of NH_3 and Cl_2 at different applied cell potentials.

3. Does the concentration of the trap solution significantly affect the capture of NH_3 ? Besides sulfuric acid, how do other trap solutions perform in capturing NH_3 ? It is suggested to include relevant data analysis of trap solutions for NH_3 capture to ensure the scalability of NH_3 recovery in the system.

Reply: The concentration of the trap solution has a minimal impact on the effectiveness of NH_3 capture, as the driving force for NH_3 capture is the concentration gradient or vapor pressure difference of NH_3 gas across the gas diffusion layer. Essentially, any acidic solution can serve as an effective trap for NH_3 , provided it sufficiently lowers the pH below the ammonium/ammonia dissociation equilibrium point ($\text{pK}_a = 9.24$).¹ The critical factor to consider is the solution's capacity to consume H^+ ions during the NH_3 trapping process, which is determined by both the concentration and volume of the trap solution and the total amount

of NH_3 expected to be captured over the operation period. In our experiments with synthetic waste streams, we utilized a H_2SO_4 solution (0.05 M; pH 1). Other studies have successfully employed varying concentrations of H_2SO_4 , ranging from 0.01 M to 1 M, all demonstrating similar NH_3 trapping efficiencies.^{2,3}

According to your suggestion, we explored NH_3 capture using alternative acidic solutions such as HNO_3 , H_3PO_4 , and a mixed acid solution comprising H_2SO_4 , HNO_3 , and H_3PO_4 in a 1:1:1 molar ratio with the same initial pH of 1. Furthermore, to avoid the use of hazardous acids and bases for NH_3/Cl_2 trapping, we have piloted an electrochemical method that generates acid and base solutions directly from the electrolysis process by utilizing a 0.1 M Na_2SO_4 solution. This additional data and the comparative analysis of various trap solutions for NH_3 capture have been thoroughly documented in the "Economic Analysis" section and illustrated in Supplementary **Figures S13-S15**. These insights not only validate the versatility and scalability of NH_3 recovery in our system but also highlight the potential for efficient NH_3 trapping methods.

Line 347-359, page 22.

To produce different ammonium salts, we employed HNO_3 , H_3PO_4 , and a mixed-acid solution comprising HNO_3 , H_2SO_4 , and H_3PO_4 in a 1:1:1 molar ratio for NH_3 capture. The results in Supplementary Fig. S13 reveal comparable NH_3 -capture efficiencies across different acid solutions, underscoring the system's adaptability and flexibility in producing various ammonium salts. To avoid the use of hazardous chemicals, in-situ acid/alkaline production via water electrolysis reaction was achieved with a proton exchange membrane-separated electrolyzer, featuring a stainless-steel mesh cathode and DSA anode, which generated the acid and alkaline solutions from a 0.1 M Na_2SO_4 feed. As shown in Supplementary Fig. S14, two types of operations were tested: two-stage (where acid/alkaline solutions are generated first and then used for NH_3/Cl_2 capture) and single-stage (where acid-base solutions are generated simultaneously while capturing NH_3/Cl_2). Although NH_3 capture efficiency remained consistent, significant Cl_2 product loss (85%) was observed in the single-stage operation (Supplementary Fig. 15). This loss is likely because NH_4^+ cannot be readily oxidized at the

anode, whereas OCI^- can be reduced to Cl^- at the cathode.^{16,38}

Figure S13. The comparison of various acid solutions for NH_3 capture. Specific testing conditions: electrolyte channel: 25 mM NaNO_3 , 25 mM NaCl , 0.1 M Na_2SO_4 , pH 7.0; cell voltage 3.0 V. The flow rates of all electrolytes are $25 \text{ mL}\cdot\text{min}^{-1}$.

Figure S14. (a) two-stage operation, where the acid/base was generated in a proton exchange membrane (PEM)-separated electrolyzer and then pumped into another electrochemical reactor for NH_3/Cl_2 separation (batch mode). (b) one-stage operation, where the acid/base was generated and continuously pumped into the electrochemical reactor for NH_3/Cl_2 separation (continuous mode).

Figure S15. The comparison of two-stage operation and one-stage operation for NH₃ (a) and Cl₂ (b) capture. Specific testing conditions: electrolyte channel: 25 mM NaNO₃, 25 mM NaCl, 0.1 M Na₂SO₄, pH 7.0; cell voltage 3.0 V. The flow rates of all electrolytes are 25 mL·min⁻¹.

References cited in this response letter:

1. Dantie, Mekdimu Mezemir, et al. "Ammonia recovery from human urine as liquid fertilizers in hollow fiber membrane contactor: Effects of permeate chemistry." *Environmental Engineering Research* (2020).
2. Iddya, Arpita, et al. "Efficient ammonia recovery from wastewater using electrically conducting gas stripping membranes." *Environmental Science: Nano* 7.6 (2020): 1759-1771.
3. Tarpeh, William A., et al. "Electrochemical stripping to recover nitrogen from source-separated urine." *Environmental science & technology* 52.3 (2018): 1453-1460.

4. In line 192, the author mentions the main reactions occurring in the electrolytic cell, including the reduction of HClO. The preceding section has demonstrated the transformation of hypochlorite into NaClO₄ in the trap solution. Please confirm the chemical forms involved in this part of the study. Additionally, in line 203, the author states, "N₂ was dominant because the reaction between NH₃/NH₄⁺ and HClO/ClO⁻ generates N₂ (4.2×10⁶ M⁻¹·s⁻¹) faster than NO₃⁻ (0.1–0.7 M⁻¹·s⁻¹)." During this process, HClO cannot react with NO₃⁻, so it is recommended to revise the narrative in this section.

Reply: (1) In response to the first query, within the electrolyzer, before NH₃ and Cl₂ are separated, they may participate in various redox processes. Upon separation, NH₃ and Cl₂ are chemically transformed into (NH₄)₂SO₄ and NaClO, respectively, in their specific trap

solutions. Supplementary Table 1 summarized the major reactions that may take place within the electrolyzer without the separation of NH₃ and Cl₂.

(2) Addressing the second concern, we first analyzed the interactions between nitrogen and chlorine species within the electrolyzer without the separation of NH₃ and/or Cl₂. The formation of N₂, NO₃⁻, and Cl⁻ as final products stems from the reactions between NH₃/NH₄⁺ and HClO/ClO⁻ species. This part serves as a control experiment to verify whether the separation process of NH₃/Cl₂ is fast enough to outcompete their redox reaction. We revised this section and restructured and supplied Supplementary Table 1 as follows:

Line 198-212, page 13:

To gain insights into the effect of NH₃/Cl₂ separation on reducing product loss, we conducted a series of control experiments by varying the concentrations/ratios of nitrogen and chloride species in the feed electrolyte, with and without the incorporation of separation operations. The major heterogeneous and homogeneous redox reactions within the electrolyzer are shown in **Fig. 3a** and summarized in Supplementary Table 1.³⁸ As shown in **Fig. 3b** and **c**, the introduction of Cl⁻ ions resulted in an observable increase in the remaining NO₃⁻ concentration (blue data points), from 7.3±0.2 mM to 9.5±0.2 mM, and a corresponding decrease in the final NH₃ (purple data points) concentration, from 13.4±0.1 mM to 10.4±0.5 mM. The concentration of N₂ (green data points), encompassing both dissolved and vaporized forms, increased from 2.0±0.4 mM to 4.6±0.3 mM. This calculation was based on the disparity between the input nitrate nitrogen and the nitrogen species retained in the solution. The amplified N₂ concentration is attributed to the rapid reaction kinetics towards N₂ (4.2×10⁶ M⁻¹·s⁻¹) compared to NO₃⁻ (0.1–0.7 M⁻¹·s⁻¹) in the context of NH₃/NH₄⁺ interaction with HClO/ClO⁻.^{26,37} As the rate-limiting species for NO₃⁻ reduction, the NO₂⁻ concentration (gray data points) reduced from 2.3±0.1 mM to 0.6±0.1 mM, when Cl⁻ was present.

Supplementary Table 1. Homogeneous/heterogeneous redox reactions related to nitrogen and chlorine species within the electrolyzer.

No.	Reactions	Rate constants or reaction potentials	Refs
Nitrate reduction reactions			
NO_3^- to NO_2^-			
1	$NO_3^-_{(ad)} + e^- \rightarrow NO_3^{2-}_{(ad)}$	$E^0 = -0.89$ V vs SHE	4
2	$NO_3^{2-}_{(ad)} + H_2O \rightarrow NO_2^{\bullet}_{(ad)} + 2OH^-$	$k = 5.5 \times 10^4$ s ⁻¹	5
3	$NO_2^{\bullet}_{(ad)} + e^- \rightarrow NO_2^-_{(ad)} + H_2O$	$E^0 = 1.04$ V vs SHE	6
NO_2^- to NO			
4	$NO_2^-_{(ad)} + e^- \rightarrow NO_2^{2-}_{(ad)}$	$E^0 = -0.47$ V vs SHE	7
5	$NO_2^{2-}_{(ad)} + H_2O \rightarrow NO_{(ad)} + 2OH^-$	$k = 1.0 \times 10^5$ s ⁻¹	8
NO to N_2			
Pathway 1			
6	$NO_{(ad)} + 2H^+ + 2e^- \rightarrow N_{(ad)} + H_2O$	E^0 is unavailable	9
7	$N_{(ad)} + N_{(ad)} \rightarrow N_2$	N.A.	9
Pathway 2 (Vooyo-Koper mechanism)			
8	$NO_{(ad)} + NO_{(aq)} + H^+ + e^- \rightarrow HN_2O_2$	$E^0 = 0.00$ V vs SHE	10
9	$HN_2O_{2(ad)} + H^+ + e^- \rightarrow N_2O_{(ad)} + H_2O$	$E^0 = 1.59$ V vs SHE	11
10	$N_2O + e^- \rightarrow N_2O^-$	$E^0 = 1.77$ V vs SHE	12
11	$N_2O^- + 2H^+ + e^- \rightarrow N_2 + H_2O$	N.A.	13
Pathway 3 (Duca-Feliu-Koper mechanism)			
12	$NO_{(ad)} + 3H_2O + 4e^- \rightarrow NH_{2(ad)} + 4OH^-$	E^0 is unavailable	14
13	$NO_{(ad)} + NH_{2(ad)} \rightarrow NONH_{2(ad)}$	N.A.	15
14	$NONH_{2(ad)} \rightarrow N_2 + H_2O$	N.A.	15
NO to NH_3			
15	$O_{(ad)} + H^+ + e^- \rightarrow HNO_{(ad)}$	$E^0 = -0.78$ V vs SHE	16
16	$HNO_{(ad)} + H^+ + e^- \rightarrow H_2NO_{(ad)}$	$E^0 = 0.52$ V vs SHE	9
17	$H_2NO_{(ad)} + H^+ + e^- \rightarrow H_2NOH_{(ad)}$	$E^0 = 0.90$ V vs SHE	17
18	$H_2NO_{(aq)} + 2H^+ + 2e^- \rightarrow NH_3 + H_2O$	$E^0 = 0.42$ V vs SHE	18
19	$NH_3 + H^+ \rightarrow NH_4^+$	pKa = 9.25	19
Direct nitrogen species oxidation reaction			

20	$NH_3 + 3OH^- \rightarrow \frac{1}{2}N_2 + 3H_2O + 3e^-$	$E^0 = -0.77 \text{ V vs SHE}$	20
Chloride oxidation reactions			
21	$MO_x + Cl^- \rightarrow MO_x(Cl^*) + e^-$ (Volmer step)	$1 \text{ mA}\cdot\text{cm}^{-2}: 0.2 \times 10^{-4} \text{ s}^{-1}$ $3 \text{ mA}\cdot\text{cm}^{-2}: 1.3 \times 10^{-4} \text{ s}^{-1}$	21
22	$MO_x(Cl^*) + Cl^- \rightarrow MO_x + Cl_2 + e^-$ (Heyrovsky)	$5 \text{ mA}\cdot\text{cm}^{-2}: 2.3 \times 10^{-4} \text{ s}^{-1}$ $>1.0 \times 10^5 \text{ M}^{-1}\cdot\text{s}^{-1}$	21
23	$MO_x(Cl^*) + MO_x(Cl^*) \rightarrow 2MO_x + Cl_2$	$1.0 \times 10^8 \text{ M}^{-1}\cdot\text{s}^{-1}$	22
24	$Cl_2 + H_2O \rightarrow HOCl + H^+ + Cl^-$	$0.52 \text{ M}^{-1}\cdot\text{s}^{-1}$	23
25	$HOCl \leftrightarrow ClO^- + H^+$	$\text{pKa} = 7.5$	24
Chloride species reduction reactions			
26	$HOCl + 2e^- \rightarrow Cl^- + OH^-$	$1 \text{ mA}\cdot\text{cm}^{-2}: 1.8 \times 10^{-4} \text{ s}^{-1}$ $3 \text{ mA}\cdot\text{cm}^{-2}: 7.5 \times 10^{-4} \text{ s}^{-1}$ $5 \text{ mA}\cdot\text{cm}^{-2}: 11 \times 10^{-4} \text{ s}^{-1}$	21
Interactions between nitrogen/chloride species			
HOCl mediated ammonia oxidation			
27	$NH_3 + HOCl \rightarrow NH_2Cl + H_2O$	$3.1 \times 10^6 \text{ M}^{-1}\cdot\text{s}^{-1}$	25
28	$NH_2Cl + HOCl \rightarrow NHCl_2 + H_2O$	$1.5 \times 10^2 \text{ M}^{-1}\cdot\text{s}^{-1}$	26
29	$NHCl_2 + H_2O \rightarrow NOH + 2H^+ + 2Cl^-$	$1.7 \times 10^2 \text{ M}^{-1}\cdot\text{s}^{-1}$	27
30	$NH_2Cl + NOH \rightarrow N_2 + H^+ + Cl^- + H_2O$	$8.3 \times 10^3 \text{ M}^{-1}\cdot\text{s}^{-1}$	27
31	$NHCl_2 + NOH \rightarrow N_2 + 2H^+ + 2Cl^- + H_2O$	$2.8 \times 10^4 \text{ M}^{-1}\cdot\text{s}^{-1}$	27
32	$NHCl_2 + 2HOCl + H_2O \rightarrow NO_3^- + 5H^+ + 4Cl^-$	$2.3 \times 10^2 \text{ M}^{-1}\cdot\text{s}^{-1}$	28
33	$NH_3 + 4HOCl \rightarrow NO_3^- + 3H_2O + H^+ + 4Cl^-$	$0.1-0.7 \text{ M}^{-1}\cdot\text{s}^{-1}$	21
HOCl/NH₂Cl mediated nitrite oxidation			
34	$NO_2^- + HOCl \rightarrow NO_2Cl + OH^-$	$1.8 \times 10^5 \cdot \text{M}^{-1}\cdot\text{s}^{-1}$	29
35	$H^+ + NH_2Cl + NO_2^- \rightarrow NH_3 + NO_2Cl$	unknown	
36	$NO_2Cl + NO_2^- \rightarrow N_2O_4 + Cl^-$	unknown	
37	$N_2O_4 + OH^- \rightarrow NO_3^- + NO_2^- + H^+$	fast	
38	$NO_2Cl \rightarrow NO_2^+ + Cl^-$	unknown	
39	$NO_2^+ + OH^- \rightarrow NO_3^- + H^+$	fast	

5. In Figure 3c, the author posits that active chlorine oxidizes NH_3 to N_2 , thereby influencing the conversion of NO_3^- and NO_2^- to NH_3 . This conclusion is contentious because, during the process, the concentration of NO_2^- rapidly decreases, transforming into a large quantity of NO_3^- and a small amount of NH_3 . Moreover, the rate of increase in NO_3^- concentration is significantly higher than the chlorination rate of NH_3 . Therefore, it is suggested that the primary reason influencing the nitrogen forms is likely the oxidation of NO_2^- to NO_3^- . It is recommended that the author compare the reaction kinetics parameters for the conversion of different nitrogen forms to elucidate the dominant reactions in the complex system.

Reply: We agree and rewrote this part according to Reviewer's comment.

Line 213-233, page 14:

Subsequent experiments aimed to trace the NO_2^- conversion pathways (e.g., conversion to NO_3^- or NH_3) and to ascertain the oxidation priorities of active chlorine species with NO_2^- and NH_3 . To this end, equal amount of NO_2^- and NH_3 were introduced together to the electrolyte, and the evolution of subsequent nitrogen species was monitored. **Fig. 3d** indicates that, during the initial 3 h, the NO_2^- concentration decreased from 12.5 mM to 1.2 ± 0.1 mM and was mainly oxidized to NO_3^- that increased from 0 mM to 8.4 ± 0.1 mM. Meanwhile, the N_2 and NH_3 concentrations increased from 0 mM and 12.5 mM to 1.2 ± 0.1 mM and 14.3 ± 0.4 mM, respectively. After 3 h when NO_2^- was nearly depleted, NH_3 oxidation became dominant as indicated by the reduced NH_3 concentration from 14.3 ± 0.4 mM to 11.3 ± 0.4 mM and the increased N_2 concentration from 1.2 ± 0.1 mM to 4.5 ± 0.7 mM. From a reaction kinetic standpoint, NH_3 has multi-step conversions with rate constants spanning from 1.7×10^2 to $3.1 \times 10^6 \text{ M}^{-1} \cdot \text{s}^{-1}$ and is considerably more vulnerable to HOCl induced oxidation than NO_2^- that has multi-step conversions with measured rate constant for only one step by far ($1.8 \times 10^5 \text{ M}^{-1} \cdot \text{s}^{-1}$).³⁹⁻⁴² The preferential reaction of NO_2^- with HOCl can be explained by the breakpoint chlorination mechanism in NH_3 , which occurs when the HOCl to NH_3 mole ratio gradually reaches 1.5.³⁷ The continuous consumption of HOCl by NO_2^- prevents the system from reaching this breakpoint chlorination ratio. During HOCl-mediated NH_3 oxidation to N_2 and NO_3^- , chloramine intermediates react with NO_2^- , converting back to NH_3 as end products.⁴¹ This process further influences the dynamics of NH_3 /HOCl interactions. The main interference

arises from HOCl that converts NO_2^- to NO_3^- at a kinetic rate five orders of magnitude faster than chloramine reactions due to the low HOCl concentration condition within the electrolyzer.⁴³

Fig. 3 | **a**, The major heterogeneous and homogeneous redox reactions within the electrolyzer. **b-f**, the nitrogen species and active chlorine species evolution over reaction time when the electrolytes in the reaction chamber contained (b) 25 mM NO_3^- ; (c) 25 mM NO_3^- , 25 mM Cl^- ; (d) 12.5 mM NO_2^- , 12.5 mM NH_3 , 25 mM Cl^- , pH 12; (e) 25 mM NO_3^- , 25 mM Cl^- ; (f) 25 mM NO_3^- , 25 mM Cl^- . All experiments were carried out under a cell potential of 3.0 V.

Added reference in the manuscript:

39. Qiang, Z. & Adams, C. D. Determination of Monochloramine Formation Rate Constants with Stopped-Flow Spectrophotometry. *Environ. Sci. Technol.* **38**, 1435-1444, (2004).
40. Margerum, D. W., Gray, E. T. J. & Huffman, R. P. in *Organometals and Organometalloids* Vol. 82 *ACS Symposium Series* Ch. **17**, 278-291 (AMERICAN CHEMICAL SOCIETY, 1979).
41. Zhang, J. *et al.* in *Recent Advances in Disinfection By-Products* Vol. 1190 *ACS Symposium Series* Ch. **5**, 79-95 (American Chemical Society, 2015).
42. Wahman, D. G. & Speitel, G. E., Jr. Relative Importance of Nitrite Oxidation by Hypochlorous Acid under Chloramination Conditions. *Environ. Sci. Technol.* **46**, 6056-6064, (2012).
43. Margerum, D. W., Schurter, L. M., Hobson, J. & Moore, E. E. Water chlorination chemistry: nonmetal redox kinetics of chloramine and nitrite ion. *Environ. Sci. Technol.* **28**, 331-337 (1994).

6. In lines 255-257, the author suggests that the higher concentrations of NO_3^- and Cl^- result in increased NH_3/Cl_2 yields, indicating that mass transfer on the gas diffusion layer is not a limiting factor. However, this assertion needs further confirmation, as the author only explores the concentration increase from 10 mM to 100 mM for NO_3^- and Cl^- . This range may not adequately reveal the limitations imposed by mass transfer on the increase in yields. Alternatively, the author may consider citing references that provide evidence for the typical concentrations of NO_3^- and Cl^- in concentrated water within this specified range.

Reply: The chosen concentration ranges are reflective of real-world scenarios: NO_3^- and Cl^- concentrations around 10 mM are typical in industrial runoff,¹ while concentrations up to 100 mM are often found in brine wastewater from processes like ion exchange or reverse osmosis². For instance, the reverse osmosis retentate from the Yuma Desalination Plant in Arizona, used in our study, shows NO_3^- and Cl^- concentrations of approximately 11.8 mM and 54.1 mM, respectively, as listed in Supplementary Table 2. This context underscores the practical applicability of our findings across the specified concentration ranges. We further modified the relevant description based on the reviewer's suggestion as below.

Table S2. The characteristics of the obtained reverse osmosis (RO) retentate obtained from a RO plant for surface and ground water treatment.

Component	Concentration (mg L ⁻¹)
Conductivity	9790 uS cm ⁻¹
pH	6.57 (measured at 19.5°C)
Nitrate as N	164.7 (~11.8 mM)
Aluminum	0.07
Barium	0.04
Bicarbonate as CaCO ₃	11.3
Boron	1.47
Calcium as CaCO ₃	775
Chloride	1920 (~54.1 mM)
Chromium	0.15
Fluoride	<20
Iron	1.53
Magnesium as CaCO ₃	650
PO ₄ ⁻ as P	<20
Potassium	32.1
Silicon Dioxide	26.0
Sodium	1840
Strontium	4.24
Sulfate	2680

References cited in this response letter:

1. Zheng, Wenxiao, et al. "Self-activated Ni cathode for electrocatalytic nitrate reduction to ammonia: from fundamentals to scale-up for treatment of industrial wastewater." *Environmental Science & Technology* 55.19 (2021): 13231-13243.
2. Huo, Xiangchen, et al. "A hybrid catalytic hydrogenation/membrane distillation process for nitrogen resource recovery from nitrate-contaminated waste ion exchange brine." *Water Research* 175 (2020): 115688.

Line 268-278, page 18:

We first increased the same molar concentrations of NO₃⁻ and Cl⁻ in the feed solution from 10 mM (a typical level for most industrial wastewater⁴⁴) to 100 mM found in brine wastewater from ion exchange or reverse osmosis processes,²¹ under a fixed cell potential of 3.0 V. The left side of **Fig 4a** shows the high concentration of NO₃⁻ and Cl⁻ led to a fast reaction kinetics of NO₃⁻-to-NH₃ and Cl⁻-to-Cl₂ conversions and thus decreased the overall energy consumption from 11.9±0.1 to 6.3±0.1 kWh·kg⁻¹-products, accompanied by the increased NH₃/Cl₂ average recovery efficiency (95% to 98%) and two product generation rates or fluxes (373±7 to

1763±25 g-(NH₄)₂SO₄·m⁻²·d⁻¹ and 30±3 to 625±14 g-NaClO·m⁻²·d⁻¹, respectively). This result indicates that the NH₃/Cl₂ transfer across the gas-diffusion layer was not a limiting factor and should be higher than the gaseous product generation rate on the catalyst layer under the NO₃⁻/Cl⁻ concentration in common waste streams.

7. It is recommended that the author conduct a cost assessment of the practical reverse osmosis retentate recovery system. This evaluation will help demonstrate whether the obtained values of (NH₄)₂SO₄ and NaClO surpass the total costs of trap solution and energy consumption, thereby emphasizing the relevant advantages of resource recovery in this system.

Reply: We agree, and relevant discussion was added to the manuscript.

Line 340-346, page 22.

Based on the laboratory-scale data and after deducting the associated electricity costs, the average profit attainable per metric ton for the obtained (NH₄)₂SO₄ and NaClO from synthetic waste streams with variable water chemistry parameters is projected to be \$1550, as depicted in Supplementary Fig. 12. For the real RO retentate waste stream, the projected profit is estimated at \$2364 when scaled up to an industrial scale electrolyzer. This financial metric underscores the potential economic viability of this electrosynthesis process from the waste stream.

8. *The author's exploration of the mechanism appears to be insufficiently in-depth, focusing more on quantifying the yields and recovery efficiencies of NH₃ and Cl₂. Additionally, the study lacks an extended investigation into the practical applications of the recovered NH₃ and Cl₂ in this system. While we acknowledge the engineering application value of this work, the research content seems somewhat one-sided. To enhance the study, it is suggested that the author consider incorporating economic feasibility calculations or proposing improved methods to prevent reactions between NH₃ and Cl₂, thereby further increasing the overall yield. This expansion would contribute to elevating the significance of the work.*

Reply: Thank you for your valuable feedback emphasizing the need for a deeper exploration into the mechanisms of our study and its practical applications. Here's how we have addressed these points:

(1) Prevention of NH₃ and Cl₂ Reactions: As detailed earlier (response to comment 2), we analyzed the product loss resulting from NH₃ and Cl₂ reactions. Our results demonstrate that the gas extraction electrodes effectively minimize interaction between NH₃ and Cl₂, reducing product loss to about 5%.

(2) Expansion of Capture Methods and Product Diversity: In response to comment 3, we conducted additional experiments to diversify the capture methods for NH₃/Cl₂ and explore a wider range of product applications.

(3) Mechanism Analysis: Following your suggestion (comment 4-5), we enhanced our mechanism analysis to detail how the gas extraction electrode contributes to reducing product loss, providing a clearer understanding of the underlying processes.

(4) Economic Feasibility Analysis: Addressing your seventh comment, we incorporated an economic feasibility analysis. Detailed calculations are presented in Supplementary **Figures 11-12**, which supports practical applications and the overall value and significance of our work.

Figure S11. Production cost of $(\text{NH}_4)_2\text{SO}_4$ (a) and NaClO (b) as a function of energy-related parameters and unit electricity cost.

Figure S12. The market price of obtained products under 1000 USD electricity input with a set base electricity cost of 5 ¢/kWh. The calculation is based on the recovered $(\text{NH}_4)_2\text{SO}_4$ and NaClO in the NH_3/Cl_2 trap channels.

9. The author should include information about the pH variations during the system reaction process, as side reactions can significantly impact the electrolyte pH. These variations may lead to changes in the trapping solution's performance in capturing gases. Additionally, the separation of NH_3 and Cl_2 recovered from concentrated water and the trap solution should be considered in subsequent investigations to ensure the regeneration and sustainable use of the system.

Reply: Thank you for highlighting the importance of monitoring pH variations and the regeneration and sustainability aspects of our system.

(1) pH Variations: We've documented the pH changes in the Cl_2 trap solution, NH_3 trap solution, and waste stream throughout the reaction process in **Figure S7**. In Line 172-173, page 12, we added relevant discussions:

The typical I-t curve and pH variations were illustrated in Supplementary **Fig. S7**, confirming the stability of the system.

Figure S7. (b) pH variations of Cl_2 trap solution, waste stream, and NH_3 trap solution under a cell potential of 3.0 V.

(2) System Regeneration and Sustainability: Our system's design simplifies operations by eliminating solvent-removal processes. We utilize two solutions: sulfuric acid (H_2SO_4) at pH 1 and sodium hydroxide (NaOH) at pH 13, to effectively trap ammonia (NH_3) and chlorine

gas (Cl_2), respectively. As shown in the updated **Figure 1a**, the resulting products, ammonium sulfate ($(\text{NH}_4)_2\text{SO}_4$) and sodium hypochlorite (NaClO), are directly usable as liquid fertilizer and disinfectant, avoiding further downstream processing.

Fig. 1 | Concept and verification of synergistic electrochemical synthesis and separation of NH_3 and Cl_2 . **a**, Schematic of the electrochemical NH_3 and Cl_2 production under ambient conditions using renewable energy and waste stream.

(3) Expanding product capture and conversion: To enhance product diversify the resulting liquid products, we have:

- Explored NH_3 capture using various acidic solutions such as nitric acid (HNO_3), phosphoric acid (H_3PO_4), and a mixed acid solution (H_2SO_4 , HNO_3 , H_3PO_4 in a 1:1:1 molar ratio) with the same initial pH of 1.
- Generated acidic and basic solutions for product trapping via an electrolysis process using a 0.1 M sodium sulfate (Na_2SO_4) solution without using commercial acid or base to save costs in chemicals and transport.

These updates and the corresponding data are thoroughly discussed in the "Economic Analysis and Operation Viability" section of the manuscript and detailed further in **Supplementary Figures 13-15**. These modifications ensure a comprehensive understanding of our system's functionality and its potential for sustainable industrial application.

Line 347-359, page 22.

To produce different ammonium salts, we employed HNO₃, H₃PO₄, and a mixed-acid solution comprising HNO₃, H₂SO₄, and H₃PO₄ in a 1:1:1 molar ratio for NH₃ capture. The results in Supplementary Fig. S13 reveal comparable NH₃-capture efficiencies across different acid solutions, underscoring the system's adaptability and flexibility in producing various ammonium salts. To avoid the use of hazardous chemicals, in-situ acid/alkaline production via water electrolysis reaction was achieved with a proton exchange membrane-separated electrolyzer, featuring a stainless-steel mesh cathode and DSA anode, which generated the acid and alkaline solutions from a 0.1 M Na₂SO₄ feed. As shown in Supplementary Fig. S14, two types of operations were tested: two-stage (where acid/alkaline solutions are generated first and then used for NH₃/Cl₂ capture) and single-stage (where acid-base solutions are generated simultaneously while capturing NH₃/Cl₂). Although NH₃ capture efficiency remained consistent, significant Cl₂ product loss (85%) was observed in the single-stage operation (Supplementary Fig. 15). This loss is likely because NH₄⁺ cannot be readily oxidized at the anode, whereas OCl⁻ can be reduced to Cl⁻ at the cathode.^{16, 38}

Figure S13. The comparison of various acid solutions for NH₃ capture. Specific testing conditions: electrolyte channel: 25 mM NaNO₃, 25 mM NaCl, 0.1 M Na₂SO₄, pH 7.0; cell voltage 3.0 V. The flow rates of all electrolytes are 25 mL·min⁻¹.

Figure S14. (a) two-stage operation, where the acid/base was generated in a proton exchange membrane (PEM)-separated electrolyzer and then pumped into another electrochemical reactor for NH_3/Cl_2 separation (batch mode). (b) one-stage operation, where the acid/base was generated and continuously pumped into the electrochemical reactor for NH_3/Cl_2 separation (continuous mode).

Figure S15. The comparison of two-stage operation and one-stage operation for NH_3 (a) and Cl_2 (b) capture. Specific testing conditions: electrolyte channel: 25 mM NaNO_3 , 25 mM NaCl , 0.1 M Na_2SO_4 , pH 7.0; cell voltage 3.0 V. The flow rates of all electrolytes are $25 \text{ mL} \cdot \text{min}^{-1}$.

10. The schematic diagram for the TOC requires thorough revision. The internal reaction processes within the green background are not clearly displayed, and the diagram does not effectively serve its purpose of providing an overview and introduction to the content of the article.

Reply: The schematic diagram in the Table of Contents has been totally revised to better illustrate the production of liquid fertilizer and disinfectant from waste streams using a membrane-free electrolyzer. We hope the new design provides a clearer overview of the article's content.

Response to Reviewer #3

The paper, 'Direct electrosynthesis and separation of ammonia and chlorine from waste streams via a stacked membrane-free electrolyzer' describes an interesting waste-water electrolysis setup with nitrate reduction occurring on the cathode and chlorine evolution on the anode. The authors measure ammonia yield/separation and chlorine yield/separation, as well as side products and reactions. Further, the system was then scaled up to a 300 cm² stack. The system is thoroughly investigated for different inlet concentrations of NO₃⁻ and Cl⁻ and electrode voltage over a number of hours and the conclusions are supported by the results. However, upon reviewing the manuscript there are a few confusingly written experimental descriptions and questions/topics raised:

Reply: Thanks for the comments. All the suggestions have been taken into account when revising this manuscript, as detailed below.

Major comments:

1. There are many illustrations and schematics in the paper, such as Fig1a, Fig1b Fig2a, Fig4c, but I did not find these useful in understanding the electrolyzer setup as Fig1a did not have enough information for the reader, Fig1b is awkwardly split into two, Fig2a mentions a 'cathodic membrane' and 'anodic membrane' when this is a membrane-less device. The most useful figure, in my opinion, is Figure S4 and this should be in the main paper, provided more detail is added to this too. For instance, 'cathodic aqueous gas extraction electrode' is too vague, it would be helpful to have the substrate, catalyst etc given on here. This looks like three layers but I can only identify two from the experimental section, being the catalyst and PTFE, what is the third layer?

Reply: We have made several revisions to address your concerns and enhance the clarity of our figures:

(1) We removed **Fig. 1a** and **1b** from the manuscript. In their place, we revised **Figure S4**, incorporating additional details, and relocated it as the new **Fig. 1a** in the main paper.

(2) In **Fig. 2a**, we corrected the terminology from “cathodic membrane” and “anodic membrane” to “cathode module” and “anode module,” respectively, to accurately reflect the membrane-less design of our device.

(3) Regarding the components of the electrode, considering you asked a similar question in comment 2, we expanded our description in response to comment 2.

Fig. 1 | Concept and verification of synergistic electrosynthesis and separation of NH_3 and Cl_2 . a, Schematic of the electrochemical NH_3 and Cl_2 production under ambient conditions using renewable energy and waste stream.

Fig. 2 | The performance of synchronous electrosynthesis and separation of NH_3 and Cl_2 of the flow-type membrane-free electrolyzer. a, Schematics and configuration of this flow-type membrane-free electrolyzer for electrochemical synthesis and *in-situ* recovery of ammonium sulfate and hypochlorous acid from waste streams.

2. There needs to be substantially more information given in the paper about how the gas-liquid separation substrates work and their materials/properties, is this a simple hydrophobic PTFE substrate or is it an AvCarb GDL? How is electronic conductivity ensured through the PTFE to the endplate?

Reply: The gas-liquid separation substrate utilized in our research is a commercial carbon-based Gas Diffusion Layer (GDL) or specifically AvCarb GDS2230. This GDL comprises a PTFE-treated carbon fiber layer, which selectively blocks liquids while permitting gas passage, and a carbon-based micro-porous layer that facilitates catalyst loading. To enhance the multilayer substrate's long-term stability, we applied an additional non-conductive hydrophobic PTFE layer atop the PTFE-treated carbon fiber layer, upon which the catalyst layer was subsequently deposited as **Figure S2** shows.

Addressing electronic conductivity, a conductive titanium sheet is affixed to the micro-porous layer side, opposite the PTFE layer. This arrangement ensures direct current flow through the micro-porous layer to the catalytic layer, effectively bypassing any conductivity issues associated with the PTFE layer. We have elaborated on this design and its functionality in the revised manuscript section.

Line 391-407, page 25, Section of MATERIALS AND METHODS, Fabrication of the Cu dendrite gas extraction cathode

A copper (Cu) dendrite electrocatalyst layer was deposited on a commercial carbon-based substrate (AvCarb GDS2230). The substrate consists of a carbon-fiber layer (PTFE treated) and a carbon-based micro-porous layer (Supplementary **Fig. S2**). To enhance the anti-wetting properties of the gas extraction electrode, the substrate was further coated with another non-conductive PTFE hydrophobic layer on the PTFE treated carbon fiber layer side via air-brush spray using a 10wt% PTFE solution and calcination operation. The catalyst was deposited on the carbon-based micro-porous layer via electrodeposition process in a typical three-electrode system. Briefly, a saturated calomel electrode (SCE, +0.241 V vs SHE), an IrO₂-RuO₂/Ti electrode (exposed area: 3 cm × 3 cm), and the substrate (exposed area: 3 cm × 3 cm) were used as reference electrode, counter electrode, and working electrode, respectively. The working electrode and counter electrode chambers were filled with a 0.1-M CuSO₄·5H₂O

solution (pH = 2, adjusted by 1 M H₂SO₄) and a 0.1-M Na₂SO₄ solution, which were separated with a proton-exchange membrane (Nafion 117). Before catalyst deposition, the microporous layer of the substrate was infiltrated with isopropanol to improve the substrate's surface wettability. A constant potential (-0.5 V vs SHE for 700 s) was applied to the working electrode by a CH Instruments 700E Potentiostat. After electrodeposition, the obtained Cu dendrite catalyst layer was rinsed with DI water and then dried in a 50°C vacuum for 5 h. The typical copper catalyst loading was ~2.3 mg·cm⁻².

Figure S2. The illustration of the pretreatment of substrate, catalyst coating, and connection between conductive sheet and obtained electrode.

3. There is inconsistent language which is confusing for the reader and terms are not properly defined, for instance, ‘ammonia trap channel’ vs ‘ammonia recovery channel’ seem to be used interchangeably.

Reply: We agree with your observation regarding inconsistent terminology, which may lead to confusion. To address this, we conducted a comprehensive review of the entire manuscript, ensuring that all text and figures use consistent and properly defined terms. Moving forward, we consistently use “trap channel” to refer to the specific pathways for ammonia and chlorine trapping, respectively, alongside “waste stream channel” for clarity and precision in our descriptions.

4. If separation of gas-liquid occurs in the catalyst layer, how is the pH maintained by adding H_2SO_4 and $NaOH$ to the trap channel as described in the materials section?

Reply: The separation of gas and liquid within the catalyst layer, as detailed in the Supporting Information **Part S1** "Electrode Interfacial pH Calculation," results from the distinctive pH environments created at the cathode and anode—alkaline and acidic, respectively, which facilitates the formation of gaseous NH_3 at the cathode and Cl_2 at the anode. These gases then pass through the catalyst and gas diffusion layers to reach the H_2SO_4 and $NaOH$ solutions in their respective trap channels, where they are transformed into $(NH_4)_2SO_4$ and $NaClO$.

The operation of the gas-extraction electrodes incorporates a PTFE hydrophobic reinforced gas diffusion layer, which separates the waste stream from the trap solution. Consequently, the pH gradient on two sides of the electrode is maintained. The real driven force of NH_3/Cl_2 is the different vapor pressure within the catalyst layer (high) and gas diffusion layer (low) of the electrode.

We also added the pH variations of Cl_2 trap solution, NH_3 trap solution, and waste stream during the system reaction process (**Figure S7**).

Supporting information, Part S1. Electrode Interfacial pH calculation.

Cathodic/anodic interfacial pH calculation. The cathodic and anodic reactions cause an increasing ($pH^* \geq pH^b$) and decreasing ($pH^* \leq pH^b$) interfacial pH gradient toward the electrode surface, respectively. The transport fluxes of H^+ and OH^- , represented as currents, can be derived from Fick's law and are presented as follows:

$$j_{cathodic} = (1000Fm_H 10^{-pH^b})(1 - 10^{pH^b - pH^*}) + (1000Fm_{OH} 10^{pH^b - pK_w})(10^{pH^* - pH^b} - 1) \quad (S1)$$

$$j_{anodic} = (1000Fm_{OH} 10^{b - pK_w})(1 - 10^{pH^* - pH^b}) + (1000Fm_H 10^{-pH^b})(10^{pH^b - pH^*} - 1) \quad (S2)$$

where pH^b and pH^* are the bulk solution pH and interfacial pH, pK_w is the logarithm of the ionization constant of water. j is the overall response current ($A \cdot m^{-2}$). F is the Faraday constant ($96485 C \cdot mol^{-1}$), $m_H = D_H/\delta_N$ ($m \cdot s^{-1}$), $m_{OH} = D_{OH}/\delta_N$ (m/s), D_H and D_{OH} are the diffusion coefficients of H^+ and OH^- ($m^2 \cdot s^{-1}$). The diffusion layer thickness (δ_N) of H^+ and OH^- are 24.5 and 21.5 μm as calculated by Levich equation.¹

Figure S1. The relationship between cathodic/anodic interfacial pH and response current density under different bulk solution pHs. The local pH near cathode and anode was calculated as reported elsewhere.^{2,3}

Figure S7. (b) pH variations of Cl₂ trap solution, waste stream, and NH₃ trap solution under a cell potential of 3.0 V.

5. No polarization curves are provided in the manuscript for readers to understand the voltage/current relationship, some current densities are provided in the supplementary (Figure S1) but this is not sufficient

Reply: We added the typical polarization curves of different operation modules (Figure S7 and S17). We also added the summary of average current density and NH_3/Cl_2 FE for different NO_3^- and Cl^- concentrations (from 10 mM to 100 mM) and ratios ($\text{NO}_3^-:\text{Cl}^-$ concentration ratios from 4:1 to 1:4) under the fixed cell potential of 3.0 V (Table S2).

Figure S7. (a) I-t curve under a cell potential of 3.0 V.

Figure S17. Typical I-t curves of the continuous module for NH_3 and Cl_2 production and separation under different flow rates and the same cell potential (3.0 V).

Table S2. Summary of current density and NH₃/Cl₂ FE of different NO₃⁻ and Cl⁻ concentrations and ratios under the fixed 3.0 V cell potential.

Concentrations and ratios	Current densities	NH ₃ FE	Cl ₂ FE
NO₃⁻:Cl⁻ concentration ratio=1			
10 mM NO ₃ ⁻ , 10 mM Cl ⁻	4.00±0.22 mA·cm ⁻²	69.9±1.8%	2.7±0.6%
25 mM NO ₃ ⁻ , 25 mM Cl ⁻	5.67±0.56 mA·cm ⁻²	79.1±1.1%	9.6±0.2%
50 mM NO ₃ ⁻ , 50 mM Cl ⁻	8.44±0.67 mA·cm ⁻²	87.8±2.0%	25.6±0.9%
100 mM NO ₃ ⁻ , 100 mM Cl ⁻	13.56±1.22 mA·cm ⁻²	92.6±0.5%	36.9±0.1%
NO₃⁻:Cl⁻ concentration ratios from 4:1 to 1:4			
25 mM NO ₃ ⁻ , 6.75 mM Cl ⁻	5.67±0.44 mA·cm ⁻²	74.3±0.7%	0.5±0.1%
25 mM NO ₃ ⁻ , 12.5 mM Cl ⁻	5.78±0.67 mA·cm ⁻²	75.4±0.4%	0.7±0.0%
25 mM NO ₃ ⁻ , 25 mM Cl ⁻	5.67±0.56 mA·cm ⁻²	79.1±1.1%	9.6±0.2%
25 mM NO ₃ ⁻ , 50 mM Cl ⁻	6.78±0.33 mA·cm ⁻²	63.7±2.8%	18.1±0.5%
25 mM NO ₃ ⁻ , 100 mM Cl ⁻	8.00±0.89 mA·cm ⁻²	54.7±2.5%	36.1±0.7%

6. What are your faradaic efficiencies?

Reply: As listed in our Response to major comment 5, the Faradaic Efficiencies were added.

Minor comments

1. There are some typos in the manuscript, please read and edit carefully.

Reply: We conducted a thorough review of the entire manuscript to correct any typographical errors.

2. Please also describe the 'recovery channel', is this simply a flow channel or is there a specific property to it?

Reply: As listed in response to major comment-3, we deleted all "recovery channel" and all use "trap channel". It's simply a flow channel. We also added the description in the manuscript where it first appeared.

Line 152-155, page 11

The gas-extraction electrodes were incorporated into a flow-type membrane-free electrolyzer, which consisted of an ammonia trap channel (circulating the H₂SO₄ solution), a chlorine trap channel (circulating the NaOH solution), and a waste stream channel (Fig. 2a).

The physical installation diagram of the electrolyzer is shown in Supplementary Fig. 6.

3. If H₂SO₄ and NaOH are added to the electrolyzer, how much of this exits in the 'clean' water and does this still meet the wastewater regulatory limits?

Reply: The hydrophobic PTFE reinforced gas diffusion layer and the electrocatalyst layer faced the trap solution and waste stream, respectively. The gas diffusion layer only allows the transfer of gases and repel the across of ions. According to our previous study,¹ the ions or impurities (COD, Cl⁻, Na⁺, K⁺, and SO₄²⁻) in waste stream or trap solution can barely pass through the gas diffusion layer (lower than the detection limit).

We also recognize that the effluent from our proposed system cannot be considered as "clean water" due to the presence of other ions, such as phosphate, fluoride, and potentially toxic organic pollutants, in the feed stream. This technology aims to reduce the concentrations of NO₃⁻ and Cl⁻ in the input waste stream and ensure those chemicals, such as NH₃/NH₄⁺ (0.3 mM), NO₂⁻ (0.2 mM), and Cl₂/HClO/ClO⁻ (0.1 mM), in the discharged waste stream will comply with discharge regulations for nitrogen- and chlorine-containing species. We have revised the wording in the "ABSTRACT" and "Large-scale electrosynthesis using a stacked electrolyzer system" sections to clarify this point and ensure there is no overstatement of our findings. Additionally, the annotations in Fig 2a have been updated to reflect this clarification.

Line 22-25, page 2	
Original	Revised
This yielded high concentrations of (NH ₄) ₂ SO ₄ (83.8 mM) and NaClO (243.4 mM) at an electrical cost of 7.1 kWh per kilogram of solid products, while maintaining minimal residual pollutants	This yielded high concentrations of (NH ₄) ₂ SO ₄ (83.8 mM) and NaClO (243.4 mM) at an electrical cost of 7.1 kWh per kilogram of solid products, while residual NH ₃ /NH ₄ ⁺ (0.3 mM), NO ₂ ⁻ (0.2 mM), and

(NH ₃ /NH ₄ ⁺ : 0.3 mM, NO ₂ ⁻ : 0.2 mM, Cl ₂ /HClO/ClO ⁻ : 0.1 mM) in the waste stream and meeting wastewater discharge regulations.	Cl ₂ /HClO/ClO ⁻ (0.1 mM) pollutants in the waste stream could meet the wastewater discharge regulations for nitrogen- and chlorine-species.
Line 305-309, page 20	
Original	Revised
Concurrently, the residual concentrations of NH ₃ /NH ₄ ⁺ , Cl ₂ /HClO/ClO ⁻ , and NO ₂ ⁻ in the treated reverse osmosis retentate were found at 0.32±0.19 mM, 0.06±0.02 mM, and 0.15±0.08 mM (Fig. 4e), respectively. These values are below the regulatory limits for wastewater discharges into receiving water bodies.	Concurrently, the residual concentrations of NH ₃ /NH ₄ ⁺ , Cl ₂ /HClO/ClO ⁻ , and NO ₂ ⁻ in the treated reverse osmosis retentate were found at 0.32±0.19 mM, 0.06±0.02 mM, and 0.15±0.08 mM (Fig. 4e), respectively. These values are below the relevant nitrogen- and chlorine-species regulatory limits for wastewater discharges into receiving water bodies.

Fig. 2 | The performance of synchronous electrosynthesis and separation of NH₃ and Cl₂ of the flow-type membrane-free electrolyzer. **a, Schematics and configuration of this flow-type membrane-free electrolyzer for electrochemical synthesis and *in-situ* recovery of ammonium sulfate and hypochlorous acid from waste streams.**

References cited in this response letter:

1. Gao, Jianan, et al. "Electrochemically selective ammonia extraction from nitrate by coupling electron- and phase-transfer reactions at a three-phase interface." *Environmental Science & Technology* 55.15 (2021): 10684-10694.

Response to Reviewer #4

I very much enjoyed reading this manuscript since it is not the typical performance-based catalyst development. This said, there are several things that would have to be resolved and thus I cannot support publication of this study.

Reply: Thank you for your feedback and acknowledging the unique focus of our manuscript. We are committed to addressing the issues you've identified to meet the publication standards of this study.

1. One of the biggest issue with the manuscript is that the authors show no composition analysis of the actual waste water that is used herein. This might lead to reproducibility issues since additional ions can hamper the overall process. Along this line, it is questionable how stable the performance is against other anions and how they affect the overall performance. This needs to be done however to show the robustness of the process. In addition, it is not a good sign when only in the experimental part it is hidden that the authors utilized synthetic waste water solutions. The experiments need to be performed with real life ones to show the robustness of the system. Otherwise the story is simply made up and it has no relation to any waste water treatment system.

Reply: We acknowledge the concern raised regarding the absence of a composition analysis for the actual wastewater used in our study and its potential impact on reproducibility and performance stability against various anions. We made numerous changes to remedy this situation.

First, the composition of the wastewater used in our experiments, real reverse osmosis retentate from the Yuma Desalination Plant in Arizona, is provided in **Supplementary Table 3** to ensure transparency and reproducibility. The efficacy of our stacked electrolyzer system in treating actual reverse osmosis (RO) retentate is demonstrated in **Fig. 4d** and **4e**, highlighting its applicability to real-life wastewater treatment scenarios.

Regarding the synthetic waste streams, the composition was mentioned in the manuscript (lines 120-121, page 8, “A synthetic medium-strength waste stream containing either 25 mM NO_3^- or 25 mM Cl^- were employed.”) and (lines 262-264, page 18, “Supplementary Fig. 9-10

shows the time-resolved evolution of nitrogen and chlorine species for synthetic waste stream with different nitrate and chloride concentrations or ratios).

To avoid any potential misunderstandings, we carefully reviewed the entire manuscript to emphasize the differentiation between experiments conducted using synthetic electrolytes and those using actual wastewater:

Line 167-172, page 11.

“As demonstrated in **Fig. 2b** and **2c**, after 5 hours of single-electrosynthesis with the **synthetic** waste stream of either 25 mM NO_3^- or 25 mM Cl^- , the concentrations of NH_3 and Cl_2 in their respective trap solutions reached 17.9 ± 0.5 mM and 9.4 ± 0.2 mM. Subsequently, we introduced a **synthetic** mixed waste stream containing 25 mM NO_3^- and 25 mM Cl^- to monitor the product concentrations during co-electrosynthesis of NH_3 and Cl_2 .”

Line 295-299, page 19.

The **real** reverse osmosis retentate from the Yuma Desalination Plant in Arizona was further used as the feed waste stream, containing average concentrations of NO_3^- and Cl^- at approximately 11.8 mM and 54.1 mM, respectively, alongside other co-existing contaminants such as Ca^{2+} , Mg^{2+} , Na^+ , SO_4^{2-} et al. A comprehensive analysis of the feed waste stream composition is provided in Supplementary Table 3.

Line 340-344, page 22.

Based on the laboratory-scale data and after deducting the associated electricity costs, the average profit attainable per metric ton for the obtained $(\text{NH}_4)_2\text{SO}_4$ and NaClO from **synthetic** waste streams with variable water chemistry parameters is projected to be \$1550, as depicted in Supplementary Fig. 12. For the **real** RO retentate waste stream, the projected profit is estimated at \$2364 when scaled up to an industrial scale electrolyzer.

Line 451-453, page 28, MATERIALS AND METHODS, section of Electrolyzer setup and operation.

The **real** reverse osmosis retentate from the Yuma Desalination Plant in Arizona was also used as the feed waste stream to carry out the electrosynthesis experiment. The detailed composition analysis of the actual RO retentate was listed in Supplementary **Table S3**.

Table S3. The characteristics of the obtained reverse osmosis (RO) retentate obtained from a RO plant for surface and ground water treatment.

Component	Concentration (mg L ⁻¹)
Conductivity	9790 uS cm ⁻¹
pH	6.57 (measured at 19.5°C)
Nitrate as N	164.7 (~11.8 mM)
Aluminum	0.07
Barium	0.04
Bicarbonate as CaCO ₃	11.3
Boron	1.47
Calcium	293
Calcium as CaCO ₃	775
Chloride	1920 (~54.1 mM)
Chromium	0.15
Fluoride	<20
Iron	1.53
Magnesium as CaCO ₃	650
PO ₄ ⁻ as P	<20
Potassium	32.1
Silicon Dioxide	26.0
Sodium	1840
Strontium	4.24
Sulfate	2680

Fig. 4. d and e, The performance of stacked electrolyzer system for actual reverse osmosis (RO) retentate treatment. **The detailed composition analysis of the actual RO retentate is listed in Supplementary Table 3.** Specifically, the concentration of recovery NH_3 and Cl_2 in trap solutions and residual concentrations of $\text{NH}_3/\text{NH}_4^+$, $\text{Cl}_2/\text{HClO}/\text{ClO}^-$, and NO_2^- in RO stream as a function of operation time under 3.0 V were recorded.

2. In their introduction, the authors claim that industrial wastewater has a high salinity. This is too special and is not true for all wastewater. A more specialized phrasing is necessary.

Reply: We agree and revised the text in Line 44-53 on page 4.

Today, electrolytes such as tetrahydrofuran, toluene, and inorganic salts are employed to enhance electron transfer in electrosynthesis processes.³⁻⁸ These inputs increase the costs of input materials and for treating secondary wastes. **Conversely, wastewaters that contain dissolved contaminants could be utilized as electrolytes, offering a globally abundant and underexploited resource to tap into.**⁹⁻¹¹ Approximately 2.2×10^{15} L of wastewaters, constituting 54% of total freshwater withdrawals, is generated annually across municipal, agricultural, and industrial sectors.¹² For instance, the electrocatalytic valorization of chlorinated organic water

pollutants to ethene was recently proven feasible.¹³ Minimizing the costs of input materials, avoiding secondary contaminants, and electrochemically valorizing waste elements will offset wastewater-treatment costs.^{14,15}

Added reference in the manuscript:

12. Miller, D. M. *et al.* Electrochemical Wastewater Refining: A Vision for Circular Chemical Manufacturing. *J. Am. Chem. Soc.* **145**, 19422-19439, (2023).

3. While multiple graphics/figures show error bars, these are needed for all experiments and should be displayed accordingly.

Reply: We thoroughly reviewed all figures in the manuscript and the Supporting Information to ensure that error bars are consistently displayed for all experimental data. Specifically, error bars have now been added to **Figures 1d and 1e**. Additionally, the shaded areas in **Figures 3b through 3f** represent the error bars. We have clarified this in the legends of the respective figures to ensure clear understanding and interpretation of the data variability.

Fig. 1 | Concept and verification of synergistic electrosynthesis and separation of NH₃ and Cl₂. d and e, The recovery efficiencies of yield NH₃ and Cl₂ as a function of applied potentials on cathodic and anodic electrode assemblies, respectively. Synthetic waste stream is 25 mM NO₃⁻ or 25 mM Cl⁻ mixed with 0.1 M Na₂SO₄ to simulate co-existing ions in waste stream.

Fig. 3 | b-f, the nitrogen species and active chlorine species evolution over reaction time when the electrolytes in the reaction chamber contained (b) 25 mM NO₃⁻; (c) 25 mM NO₃⁻, 25 mM Cl⁻; (d) 12.5 mM NO₂⁻, 12.5 mM NH₃, 25 mM Cl⁻, pH 12; (e) 25 mM NO₃⁻, 25 mM Cl⁻; (f) 25 mM NO₃⁻, 25 mM Cl⁻. All experiments were carried out under a cell potential of 3.0 V. **The shadow area in the figure represents the error scale.**

4. The authors mainly rely on their explanations on the analytics reported after 5 hours. This is not enough time to make a reasonable statement. As it looks like from the data, the system did not yet reach a steady state performance and thus no statement on performance should be provided. Likewise, this referee wants to know how the system behaves in single pass mode when no recirculation of the storage tanks is used. This is the more likely scenario and it is important to show the performance in a continuous flow.

Reply: Thank you for your insightful observations regarding the experimental duration and operational conditions of our system. Here's how we've addressed your concerns:

(1) Duration of Experiments: The 5-hour timeframe for the mechanism experiments illustrated in **Figure 3** was specifically chosen to align with the experiment conditions in **Figure 2** so that it would explain the effect of NH₃/Cl₂ separation on reducing product loss. We acknowledge your point about extending the reaction time; however, extending beyond 5

hours tends to lead to increased product loss, which may not be reflective of practical electrosynthesis and separation scenarios. Hence, the decision to limit the duration to 5 hours was to maintain relevance to typical operational times.

(2) Single Pass Mode Performance: In response to your second point, we conducted additional experiments to assess the system's performance in a single pass mode, where there was no recirculation of the waste stream storage tank. This setup is indeed more reflective of likely real-world applications and provides new insights into the system's efficiency in continuous flow conditions. The results of these experiments have been added to Figure S16 in the Supporting Information. A detailed discussion of these findings has also been included in the manuscript to ensure comprehensive understanding and accessibility.

Line 360-370, page 23.

We further conducted experiments in a single-pass mode without recirculating the waste stream storage tank, detailed in Supplementary Fig. S16-17. At flow rates of $5 \text{ mL}\cdot\text{min}^{-1}$ and $25 \text{ mL}\cdot\text{min}^{-1}$, the yields of $(\text{NH}_4)_2\text{SO}_4$ reached $64.51\pm 3.21 \text{ mM}$ and $23.43\pm 1.19 \text{ mM}$, respectively, after 5 hours, with NH_3 separation efficiencies of 54% and 71%. The discharged waste stream contained NO_2^- and NH_3 concentrations ranging from $0.21\pm 0.02 \text{ mM}$ to $1.07\pm 0.06 \text{ mM}$ and $0.96\pm 0.06 \text{ mM}$ to $5.48\pm 0.18 \text{ mM}$, respectively. The Cl_2 separation efficiencies exceeded 99%, maintaining residual active chlorine concentrations in the discharged waste stream below 0.01 mM throughout. To ensure compliance with regulatory limits, careful control of the flow rate is essential. These findings provide critical data for decision-makers and stakeholders to assess the economic benefits and potential applications of this membrane-free electrolyzer in chemical synthesis with waste streams.

Figure S16. The continuous module for NH₃ and Cl₂ production and separation. Specific testing conditions: electrolyte channel: 25 mM NaNO₃, 25 mM NaCl, 0.1 M Na₂SO₄, pH 7.0; cell voltage 3.0 V. The flow rates of all electrolytes are 5 mL·min⁻¹ and 25 mL·min⁻¹.

5. The authors claim that after their process they obtain clean water. I do not see any evidence for this since no detailed analysis to underline this statement is presented.

Reply: We recognize that the effluent from our proposed system cannot be considered as “clean water” due to the presence of other ions, such as phosphate, fluoride, and potentially toxic organic pollutants, in the feeding waste stream.

This technology aims to reduce the concentrations of NO₃⁻ and Cl⁻ in the input waste stream and ensure those chemicals, such as NH₃/NH₄⁺ (0.3 mM), NO₂⁻ (0.2 mM), and Cl₂/HClO/ClO⁻ (0.1 mM), in the discharged waste stream will comply with discharge regulations for nitrogen- and chlorine-containing species. We have revised the wording in the “ABSTRACT” and “Large-scale electrosynthesis using a stacked electrolyzer system” sections

to clarify this point and ensure there is no overstatement of our findings. Additionally, the annotations in **Fig 2a** have been updated to reflect this clarification.

Line 22-25, page 2	
Original	Revised
This yielded high concentrations of (NH₄)₂SO₄ (83.8 mM) and NaClO (243.4 mM) at an electrical cost of 7.1 kWh per kilogram of solid products, while maintaining minimal residual pollutants (NH₃/NH₄⁺: 0.3 mM, NO₂⁻: 0.2 mM, Cl₂/HClO/ClO⁻: 0.1 mM) in the waste stream and meeting wastewater discharge regulations.	This yielded high concentrations of (NH₄)₂SO₄ (83.8 mM) and NaClO (243.4 mM) at an electrical cost of 7.1 kWh per kilogram of solid products, while residual NH₃/NH₄⁺ (0.3 mM), NO₂⁻ (0.2 mM), and Cl₂/HClO/ClO⁻ (0.1 mM) pollutants in the waste stream could meet the wastewater discharge regulations for nitrogen- and chlorine-species.
Line 305-309, page 20	
Original	Revised
Concurrently, the residual concentrations of NH₃/NH₄⁺, Cl₂/HClO/ClO⁻, and NO₂⁻ in the treated reverse osmosis retentate were found at 0.32±0.19 mM, 0.06±0.02 mM, and 0.15±0.08 mM (Fig. 4e), respectively. These values are below the regulatory limits for wastewater discharges into receiving water bodies.	Concurrently, the residual concentrations of NH₃/NH₄⁺, Cl₂/HClO/ClO⁻, and NO₂⁻ in the treated reverse osmosis retentate were found at 0.32±0.19 mM, 0.06±0.02 mM, and 0.15±0.08 mM (Fig. 4e), respectively. These values are below the relevant nitrogen- and chlorine-species regulatory limits for wastewater discharges into receiving water bodies.

Fig. 2 | The performance of synchronous electrosynthesis and separation of NH_3 and Cl_2 of the flow-type membrane-free electrolyzer. **a, Schematics and configuration of this flow-type membrane-free electrolyzer for electrochemical synthesis and *in-situ* recovery of ammonium sulfate and hypochlorous acid from waste streams.**

6. I a missing typical U/I and U or I/t curves for all provided experiments. What is the typical current density the authors measure at? This is important to judge the energy efficiency of the system and ohmic losses.

Reply: We addressed this by adding polarization curves for different operational modules in **Figures S7** and **S17**. Additionally, we've included data on average current density and faradaic efficiency for NH_3 and Cl_2 production across various NO_3^- and Cl^- concentrations and ratios in **Table S2**.

Figure S7. (a) I-t curve under a cell potential of 3.0 V.

Figure S17. Typical I-t curves of the continuous module for NH₃ and Cl₂ production and separation under different flow rates and same cell potential (3.0 V).

Table S2. Summary of current density and NH₃/Cl₂ FE of different NO₃⁻ and Cl⁻ concentrations and ratios under the fixed 3.0 V cell potential.

Concentrations and ratios	Current densities	NH ₃ FE	Cl ₂ FE
NO₃⁻:Cl⁻ concentration ratio=1			
10 mM NO ₃ ⁻ , 10 mM Cl ⁻	4.00±0.22 mA·cm ⁻²	69.9±1.8%	2.7±0.6%
25 mM NO ₃ ⁻ , 25 mM Cl ⁻	5.67±0.56 mA·cm ⁻²	79.1±1.1%	9.6±0.2%
50 mM NO ₃ ⁻ , 50 mM Cl ⁻	8.44±0.67 mA·cm ⁻²	87.8±2.0%	25.6±0.9%
100 mM NO ₃ ⁻ , 100 mM Cl ⁻	13.56±1.22 mA·cm ⁻²	92.6±0.5%	36.9±0.1%
NO₃⁻:Cl⁻ concentration ratios from 4:1 to 1:4			
25 mM NO ₃ ⁻ , 6.75 mM Cl ⁻	5.67±0.44 mA·cm ⁻²	74.3±0.7%	0.5±0.1%
25 mM NO ₃ ⁻ , 12.5 mM Cl ⁻	5.78±0.67 mA·cm ⁻²	75.4±0.4%	0.7±0.0%
25 mM NO ₃ ⁻ , 25 mM Cl ⁻	5.67±0.56 mA·cm ⁻²	79.1±1.1%	9.6±0.2%
25 mM NO ₃ ⁻ , 50 mM Cl ⁻	6.78±0.33 mA·cm ⁻²	63.7±2.8%	18.1±0.5%
25 mM NO ₃ ⁻ , 100 mM Cl ⁻	8.00±0.89 mA·cm ⁻²	54.7±2.5%	36.1±0.7%

7. Only later on it seems that the authors used water from a desalination plant. However, also here, the composition is not specified and would require a thorough analytics to show its exact composition.

Reply: In response to comment 1, we detailed the composition analysis of the actual wastewater, specifically the real reverse osmosis retentate from the Yuma Desalination Plant in Arizona, in Supplementary Table 3. This information has now been further emphasized

within the manuscript to address concerns regarding the specificity of the water composition used in our experiments.

Line 295-299, page 19.

The real reverse osmosis retentate from the Yuma Desalination Plant in Arizona was further used as the feed waste stream, containing average concentrations of NO_3^- and Cl^- at approximately 11.8 mM and 54.1 mM, respectively, alongside other co-existing contaminants such as Ca^{2+} , Mg^{2+} , Na^+ , SO_4^{2-} et al. A comprehensive analysis of the feed waste stream composition is provided in Supplementary **Table S3**.

Line 451-453, page 28, MATERIALS AND METHODS, section of Electrolyzer setup and operation.

The real reverse osmosis retentate from the Yuma Desalination Plant in Arizona was also used as the feed waste stream to carry out the electrosynthesis experiment. The detailed composition analysis of the actual RO retentate was listed in Supplementary **Table S3**.

Table S3. The characteristics of the obtained reverse osmosis (RO) retentate obtained from a RO plant for surface and ground water treatment.

Component	Concentration (mg L^{-1})
Conductivity	9790 uS cm^{-1}
pH	6.57 (measured at 19.5°C)
Nitrate as N	164.7 (~11.8 mM)
Aluminum	0.07
Barium	0.04
Bicarbonate as CaCO_3	11.3
Boron	1.47
Calcium	293
Calcium as CaCO_3	775
Chloride	1920 (~54.1 mM)
Chromium	0.15
Fluoride	<20
Iron	1.53
Magnesium as CaCO_3	650
PO_4^- as P	<20
Potassium	32.1
Silicon Dioxide	26.0

Sodium	1840
Strontium	4.24
Sulfate	2680

Fig. 4. d and e, The performance of stacked electrolyzer system for actual reverse osmosis (RO) retentate treatment. **The detailed composition analysis of the actual RO retentate was listed in Supplementary Table 3.** Specifically, the concentration of recovery NH_3 and Cl_2 in trap solutions and residual concentrations of $\text{NH}_3/\text{NH}_4^+$, $\text{Cl}_2/\text{HClO}/\text{ClO}^-$, and NO_2^- in RO stream as a function of operation time under 3.0 V were recorded.

8. *The calculated electrical consumption of the process should be taken with care. There are no downstream processes reported herein to remove solvents etc. which can be a central part of the energy consumption. This needs to be included as an analysis.*

Reply: In response to the concern regarding the calculated electrical consumption and the potential need for downstream processes, it's important to note that our methodology does not involve the removal of solvents. Our study utilized sulfuric acid (H_2SO_4) and sodium hydroxide (NaOH) solutions to trap separated ammonia (NH_3) and chlorine gas (Cl_2), respectively. The

resultant products, ammonium sulfate ((NH₄)₂SO₄) and sodium hypochlorite (NaClO) solution, serve direct applications as liquid fertilizer and disinfectant, respectively, eliminating the need for additional downstream processing.

To broaden the versatility and scalability of NH₃ recovery within our framework, and to diversify the resultant liquid nitrogen fertilizer products, we carried out more experiments. (1) We investigated NH₃ capture using alternative acidic solutions, including nitric acid (HNO₃), phosphoric acid (H₃PO₄), and a mixed acid solution (comprising H₂SO₄, HNO₃, and H₃PO₄ in a 1:1:1 molar ratio) with the same initial pH of 1, all adjusted to the same total H⁺ concentration. (2) Moreover, in response to potential hazards associated with handling commercial acids and bases for NH₃/Cl₂ trapping, we initiated an electrochemical approach for generating acidic and basic solutions directly from the electrolysis process, utilizing a 0.1 M sodium sulfate (Na₂SO₄) solution. This novel technique further augments NH₃ and Cl₂ trapping options while mitigating safety concerns.

This additional data and comparative analysis have been meticulously outlined in the manuscript's "Economic analysis and operation viability" section and is also detailed in Supplementary **Figures S13-S15**.

Line 347-359, page 22.

To produce different ammonium salts, we employed HNO₃, H₃PO₄, and a mixed-acid solution comprising HNO₃, H₂SO₄, and H₃PO₄ in a 1:1:1 molar ratio for NH₃ capture. The results in Supplementary Fig. S13 reveal comparable NH₃-capture efficiencies across different acid solutions, underscoring the system's adaptability and flexibility in producing various ammonium salts. To avoid the use of hazardous chemicals, in-situ acid/alkaline production via water electrolysis reaction was achieved with a proton exchange membrane-separated electrolyzer, featuring a stainless-steel mesh cathode and DSA anode, which generated the acid and alkaline solutions from a 0.1 M Na₂SO₄ feed. As shown in Supplementary Fig. S14, two types of operations were tested: two-stage (where acid/alkaline solutions are generated first and then used for NH₃/Cl₂ capture) and single-stage (where acid-base solutions are generated simultaneously while capturing NH₃/Cl₂). Although NH₃ capture efficiency remained

consistent, significant Cl_2 product loss (85%) was observed in the single-stage operation (Supplementary Fig. 15). This loss is likely because NH_4^+ cannot be readily oxidized at the anode, whereas OCl^- can be reduced to Cl^- at the cathode.^{16,18}

Figure S13. The comparison of various acid solution for NH_3 capture. Specific testing conditions: electrolyte channel: 25 mM NaNO_3 , 25 mM NaCl , 0.1 M Na_2SO_4 , pH 7.0; cell voltage 3.0 V. The flow rates of all electrolytes are $25 \text{ mL} \cdot \text{min}^{-1}$.

Figure S14. (a) two-stage operation, where the acid/base was generated in a proton exchange membrane (PEM)-separated electrolyzer and then pumped into another electrochemical reactor

for NH_3/Cl_2 separation (batch mode). (b) one-stage operation, where the acid/base was generated and continuously pumped into the electrochemical reactor for NH_3/Cl_2 separation (continuous mode).

Figure S15. The comparison of two-stage operation and one-stage operation for NH_3 (a) and Cl_2 (b) capture. Specific testing conditions: electrolyte channel: 25 mM NaNO_3 , 25 mM NaCl , 0.1 M Na_2SO_4 , pH 7.0; cell voltage 3.0 V. The flow rates of all electrolytes are $25 \text{ mL}\cdot\text{min}^{-1}$.

9. Figure 4b: Why does the data shown look so wavy. It almost looks like data is added in periods? What happens when the individual rate goes down during the time of operation?

Reply: The observed waviness in the data can be attributed to the different separation efficiencies of NH_3 and Cl_2 in our system. Specifically, the NH_3 separation exhibited poorer efficiency compared to Cl_2 separation. This difference can be explained by the more varied recovery efficiencies of $(\text{NH}_4)_2\text{SO}_4$ compared to NaClO in Fig. 4b.

10. In the last sentence of the conclusion the authors finally mention a current density. This is, however misleading since no other current density is reported throughout the manuscript and it is unclear where such a number is obtained from and how it compares with the reported data.

Reply: (1) Documentation of Current Densities: We detailed the current densities of the electrolyzer for treating synthetic NO_3^- and Cl^- containing waste streams with varying concentrations (from 10 mM to 100 mM) and ratios ($\text{NO}_3^-:\text{Cl}^-$ concentration ratios from 4:1 to 1:4) in Supplementary Table S2, as referenced in response to comment 6. The average current densities recorded range from $4\text{-}14 \text{ mA}\cdot\text{cm}^{-2}$.

(2) Clarification in Conclusion: We discuss the potential for future research to explore higher current densities to align with industrial production scale demands. The exhibited low current densities when treating typical waste stream concentrations are caused by the mass transfer limitation under low waste concentration. If higher current densities are achievable, they would likely be applicable only to a limited range of specialized waste streams. To avoid any ambiguity, we have revised this portion of the conclusion to more accurately reflect the limitations and applicability concerning the current densities observed in our study.

Line 385-388, page 24

Future research ought to broaden its scope to include the electrosynthesis and separation of a more diverse array of bulk and fine chemicals employing waste stream. Additionally, it should investigate the integration of pre-concentration processes for commonly encountered low-concentration waste streams to overcome mass transfer limitations.

REVIEWER COMMENTS

Reviewer #1 (Remarks to the Author):

The authors have revised their manuscript and tried to address the reviewer's comments in terms of scientific novelty, practical performance, and cost. Although the authors have, to some extent, differentiated from existing studies in terms of reactor configuration and application scenario, there are still concerns of the novelty.

1)The authors emphasized the new concept of "reaction and separation", but it's not novel. For example, similar concept of simultaneous reduction of nitrate to ammonia and in-situ ammonia recovery was demonstrated (Environ. Sci. Technol. 2024, 58, 16, 7208–7216), in which a fluorine modified Cu electrode was developed to reduce nitrate to ammonia and an NH₃ recovery unit made from hydrophobic polypropylene membrane was adopted with H₂SO₄ as the trapping solution. In another study (Water Research, 2023, 242, 120256), a membrane free reactor was also employed with a nitrate reduction electrode, and the catholyte directly flowed through a hydrophobic polypropylene fiber membranes immersed in acid trapping solutions. In both studies, a typical reaction and separation concept was adopted, and trapping ammonia using acid was the same. In fact, the authors have previously published a very similar paper for nitrate reduction and ammonia recovery (Environ. Sci. Technol. 2021, 55, 15, 10684–10694). The authors reported using a Co based catalyst to reduce nitrate to ammonia, and the ammonia then exchanged through a hydrophobic gas transfer membrane into the trapping H₂SO₄ solution. The electrode configuration is very similar with the membrane placed next to electrode and trapping acid flowing on the other side of the membrane. The current study is just replacing the membrane with a more integrated gas diffusion layer, which is incremental and limited in science advance.

2)The authors need to clarify in which aspect (conversion efficiency, recovery efficiency, or overall cost) the proposed system outperforms existing studies in terms of ammonium sulfate or sodium hypochlorite synthesis. The nitrate part can be relatively easily compared to existing studies as previously mentioned. The chloride part is concerning as the faradic efficiency is low (oxygen evolution is dominant), and how does it compare to typical Chlor-alkali process. Specifically for brines, the sodium sulfate or sodium chloride can be crystallized and recovered in high purity. Further use of these sodium chloride for electrolysis to generate sodium hypochlorite seems more efficient and feasible.

Reviewer #2 (Remarks to the Author):

After reviewing the revisions and the responses provided by the authors, I am pleased to recommend acceptance of this manuscript. The authors have addressed the concerns raised during the review process thoughtfully and comprehensively, enhancing both the clarity and depth of the paper. The improvements made significantly contribute to the robustness and scientific merit of the work. This manuscript now presents a valuable contribution to the field and should be well-received by the

community. I look forward to seeing this work published in Nature Communication.

Reviewer #3 (Remarks to the Author):

My topics have been addressed, thank you. No further comments.

Reviewer #4 (Remarks to the Author):

the authors did a good job to answer all the questions raised by the referees and I have no doubt that this manuscript can become feasible for publication in Nature Commun. Yet, I do have some additional questions that come up during reading the revised manuscript as well as the comments that need to be answered.

1) the authors write: "We acknowledge your point about extending the reaction time; however, extending beyond 5 hours tends to lead to increased product loss, which may not be reflective of practical electrosynthesis and separation scenarios. Hence, the decision to limit the duration to 5 hours was to maintain relevance to typical operational times."

Exactly this observation is a critical point that should be reported as it has significant impact on the process conditions and subsequent optimisation strategies. Therefore, this behaviour should be shown and explained. The authors should really show the performance beyond the 5 hours.

2) The new I-t curve presented in Figure S7 reveals a small but noticeable decay over time. The authors should explain why this is the case and how it could potentially be solved.

3) In their conclusion, the authors provide a clarification on how to achieve higher current densities and argue that this is caused by mass transfer limitations under low waste concentrations. While this is partially true, the authors need to describe how to circumvent this problem since otherwise referee 1 has a valid point on the low novelty and the reactor itself renders useless for applications. Typically, the separation from mass transfer and electron transport is solved in membrane based systems by using zero-gap electrolyzers. How do the authors anticipate to make their system broadly applicable?

Response to Reviewer #1

The authors have revised their manuscript and tried to address the reviewer's comments in terms of scientific novelty, practical performance, and cost. Although the authors have, to some extent, differentiated from existing studies in terms of reactor configuration and application scenario, there are still concerns of the novelty.

Reply: Thank you for your careful evaluation of the revisions to our manuscript. We appreciate your recognition of our efforts to address the previous comments concerning scientific novelty, practical performance, and cost. We understand your concerns regarding the novelty of our study. To further clarify the unique contributions of our work, we have made a point-by-point response. We hope that these clarifications meet your expectations and strengthen the case for the manuscript's publication in Nature Communications. We are committed to ensuring our work contributes meaningfully to the advancement of our field and welcome any further suggestions you might have.

1. The authors emphasized the new concept of "reaction and separation", but it's not novel.

For example, similar concept of simultaneous reduction of nitrate to ammonia and in-situ ammonia recovery was demonstrated (Environ. Sci. Technol. 2024, 58, 16, 7208–7216), in which a fluorine modified Cu electrode was developed to reduce nitrate to ammonia and an NH₃ recovery unit made from hydrophobic polypropylene membrane was adopted with H₂SO₄ as the trapping solution. In another study (Water Research, 2023, 242, 120256), a membrane free reactor was also employed with a nitrate reduction electrode, and the catholyte directly flowed through a hydrophobic polypropylene fiber membranes immersed in acid trapping solutions. In both studies, a typical reaction and separation concept was adopted, and trapping ammonia using acid was the same.

In fact, the authors have previously published a very similar paper for nitrate reduction and ammonia recovery (Environ. Sci. Technol. 2021, 55, 15, 10684–10694). The authors reported using a Co based catalyst to reduce nitrate to ammonia, and the ammonia then exchanged through a hydrophobic gas transfer membrane into the trapping H₂SO₄ solution.

The electrode configuration is very similar with the membrane placed next to electrode and trapping acid flowing on the other side of the membrane. The current study is just replacing the membrane with a more integrated gas diffusion layer, which is incremental and limited in science advance.

Reply: Thank you for your detailed comments for the novelty of the approach to electrosynthesis and separation. We appreciate the opportunity to further clarify the distinctiveness and innovative aspects of our work in the following aspects.

- ***Novelty and System-Level Optimization:***

We would like to clarify that the scope of our study, titled "Direct Electrosynthesis and Separation of **Ammonia and Chlorine** from Waste Streams via a Stacked **Membrane-Free Electrolyzer**," encompasses more than just the synthesis and separation of ammonia. The principal innovation of our work is the integration of processes without the use of ion-exchange membranes, which we believe significantly enhances both the economic viability and scalability of the system. Unlike the references cited, our work focuses on system-level innovations that include (1) the simultaneous production of NH_3 and Cl_2 and (2) a membrane-free configuration to minimize product loss caused by spontaneous reactions between generated products. This will promote the adoption of electrosynthesis from waste streams, rather than treating it as traditional waste that require chemicals and other costs to remove, in the further.

To prove that we have discussed the **economic viability extensively** in the manuscript, showing that based on laboratory-scale data and after accounting for electricity costs, the average **profit per metric ton products** from **synthetic waste streams** (10 mM to 100 mM NO_3^- and Cl^-) is approximately **\$1550**. For **real reverse osmosis streams**, this profit escalates to **\$2364** at industrial production levels, highlighting the potential economic benefits of our system over traditional approaches.

We have gone beyond material design or electrode assembly design coupling reaction and separation. **We are not aiming to hit the highest performance value by fabricating various sophisticated catalysts.** Our strategy can be **easily extended to other electrochemical reactions or to more state-of-the-art catalysts**, include but is not limited to: (1) simultaneous

electrosynthesis and separation of products such as NH_3 , Cl_2 , and volatile fatty acids (VFA) from waste stream by higher performance catalysts; (2) electrosynthesis process need to avoid the product mixture, such as water-splitting electrolysis for H_2 and O_2 production; (3) electrosynthesis process want to **avoid the use of ion exchange membrane** to reduce the cost of electrolyzer.

- *Response to Similar Studies:*

As shown in **Figure 1** listed below, to achieve the vision about electrosynthesis based on waste streams to achieve wastewater treatment and resource recovery by electrochemical technology, we need the corporation of electrocatalyst design, electrocatalyst layer interface optimization, electrode assembly fabrication, and finally establish an electrochemical system. **To achieve this, a lot of upfront exploration into electrode assembly design is required.** While we recognize the methodologies employed in the previous studies, **our electrolyzer system uniquely integrates ammonia and chlorine production without the additional separation units like the studies you mentioned.** This integration reduces operational costs and simplifies the technology, making it more accessible and feasible for broader applications.

Figure 1. Illustration of multiscale strategies for electrosynthesis based on waste streams to solve water-energy nexus problem and achieve wastewater treatment and resource recovery.

1. Addressing Nitrogen Loss in the Presence of Chloride Ions:

Regarding the process of your concern, nitrate upcycling to ammonia and its subsequent separation, **the loss of nitrogen due to the presence of Cl^- ions in wastewater presents a significant challenge**. In the literature, whether mentioned by you or others, the prevalent approach involves **integrating an ion exchange membrane**, such as a proton exchange membrane, within the electrolyzer. This strategy is aimed at preventing the chloride evolution reaction at the anode. However, this approach raises several issues:

(1) The potential profit from the produced NH_3 exceeds the electricity costs, making the process **economically unattractive**.

(2) The **cost associated with these membranes** is considerable (account for **~24% of the electrolyzer-stack costs**), impacting the overall affordability of the system.

(3) Utilizing a membrane precludes the possibility of capturing valuable Cl_2 at the anode, thus **losing potential additional benefits**.

2. Difference from Environ. Sci. Technol. 2024, 58, 16, 7208–7216

As depicted in Figure 2, the approach outlined in the referenced paper necessitates two distinct units: (1) a wastewater treatment unit for the reduction of NO_3^- to NH_3 , accompanied by an increase in wastewater pH above 12, and (2) an NH_3 recovery unit employing a porous gas-liquid separation membrane constructed from hydrophobic polypropylene membrane fiber (PPMF). **This configuration essentially merges two separate units**—ammonia recovery often utilizes additional hollow fiber membrane modules or air purge techniques, **both of which escalate the system's cost and complexity**. In contrast, our gas extraction electrode system is designed to perform simultaneous NH_3 production and separation without the need for such supplementary units.

Moreover, it's critical to note that in the referenced study, **the financial return from recovered $(\text{NH}_4)_2\text{SO}_4$ offsets only about 15% of the electrolyzer's electricity costs, which is insufficient for profitable electrosynthesis from waste streams—a feasibility we have established in our research**. Notably, when processing real reverse osmosis (RO) streams, the profit per metric ton of product achievable in our system is approximately \$2364, significantly outperform the study you mentioned.

Figure 2. Schematic illustration of the EC–ARC system. Picture from *Environ. Sci. Technol.* 2024, 58, 16, 7208–7216.

3. Difference from *Water Research*, 2023, 242, 120256

In the study referenced, the system necessitates **two distinct units** to facilitate NH₃ generation and recovery: (1) a wastewater treatment unit, and (2) an NH₃ recovery unit, which utilizes a gas-permeable polypropylene (PP) hollow fiber membrane. **It faces the same problems as mentioned above.**

4. Difference from *Environ. Sci. Technol.* 2021, 55, 15, 10684–10694

In our previously published work, we successfully demonstrated NH₃ production and separation without the presence of Cl⁻ ions in the wastewater. However, when Cl⁻ ions are present in nitrate-containing wastewater, a proton exchange membrane (PEM) becomes essential to mitigate NH₃ loss. As shown in Figure 3, **without a PEM, NH₃ selectivity, or nitrogen product loss, escalates dramatically, decreasing by approximately 30% with 10 mM Cl⁻ and 75% with 50 mM Cl⁻.** In contrast, in this study submitted to *Nature Communications*, the total loss of produced NH₃ and Cl₂ was only 5% at a cell potential of 3.0 V.

Figure 3. The NH₃ selectivity for reduced NO₃⁻ under different Cl⁻ concentrations with/without proton exchange membrane (PEM). Figure form Environ. Sci. Technol. 2021, 55, 15, 10684–10694.

Besides, our previously published work, due to the separated design of the catalyst layer and gas diffusion layer, the ammonia recovery rate was lower than its production rate. To address this discrepancy, we introduced **an additional gas transmission tube linking the wastewater treatment chamber and the NH₃ capture chamber (Figure 4) to enhance the ammonia recovery**. However, this complex design is still less attractive than the integrated gas extraction electrode design employed in this current study that is submitted to Nature Communications.

Figure 4. The picture of the assembled electrolyzer. Figure form Environ. Sci. Technol. 2021, 55, 15, 10684–10694.

Moreover, the previous publication (Figure 5) reveals that the economic gain from producing $(\text{NH}_4)_2\text{SO}_4$ exceeds the electricity costs only when treating wastewater concentrations above 1.0 M NO_3^- , which is not common for nitrate containing waste stream. In stark contrast, this study submitted to Nature Communications (Figure S12) demonstrates that treating wastewater with $\text{NO}_3^-/\text{Cl}^-$ concentrations ranging from 10 mM to 100 mM yields an average profit of \$1550 per metric ton of products, highlighting the efficiency and economic viability of our approach.

Figure 5. Energy-related parameters and the corresponding electric energy cost as a function of NO_3^- concentration Figure form Environ. Sci. Technol. 2021, 55, 15, 10684–10694.

Figure S12. The market price of obtained products under 1000 USD electricity input.

● Response to H₂SO₄ Trapping Solution:

For the trapping solution, we also expanded to other trap solutions and trap methods rather than limited to H₂SO₄. This part was added to the part of Economic analysis and operation viability in the manuscript in Line 350-362, page 23:

To produce different ammonium salts, we employed HNO₃, H₃PO₄, and a mixed acid solution comprising HNO₃, H₂SO₄, and H₃PO₄ in a 1:1:1 molar ratio for NH₃ capture. The results in Supplementary Fig. 13 reveal comparable NH₃ capture efficiencies across different acid solutions, underscoring the system's adaptability and flexibility in producing various ammonium salts. To avoid the use of hazardous chemicals, in-situ acid/alkaline production via water electrolysis reaction was achieved with a proton exchange membrane-separated electrolyzer, featuring a stainless-steel mesh cathode and DSA anode, which generated the acid and alkaline solutions from a 0.1 M Na₂SO₄ feed. As shown in Supplementary Fig. S14, two types of operations were tested: two-stage (where acid/alkaline solutions are generated first and then used for NH₃/Cl₂ capture) and single-stage (where acid-base solutions are generated simultaneously while capturing NH₃/Cl₂). Although NH₃ capture efficiency remained consistent, significant Cl₂ product loss (85%) was observed in the single-stage operation (Supplementary Fig. 15). This loss is likely because NH₄⁺ cannot be readily oxidized at the anode, whereas OCl⁻ can be reduced to Cl⁻ at the cathode.^{16,38}

Figure S13. The comparison of various acid solution for NH₃ capture. Specific testing conditions: electrolyte channel: 25 mM NaNO₃, 25 mM NaCl, 0.1 M Na₂SO₄, pH 7.0; cell voltage 3.0 V. The flow rates of all electrolytes are 25 mL·min⁻¹.

Figure S14. (a) two-stage operation, where the acid/base was generated in a proton exchange membrane (PEM)-separated electrolyzer and then pumped into another electrochemical reactor for NH_3/Cl_2 separation (batch mode). (b) one-stage operation, where the acid/base was generated and continuously pumped into the electrochemical reactor for NH_3/Cl_2 separation (continuous mode).

Figure S15. The comparison of two-stage operation and one-stage operation for NH_3 (a) and Cl_2 (b) capture. Specific testing conditions: electrolyte channel: 25 mM NaNO_3 , 25 mM NaCl , 0.1 M Na_2SO_4 , pH 7.0; cell voltage 3.0 V. The flow rates of all electrolytes are $25 \text{ mL} \cdot \text{min}^{-1}$.

- **Broader Implications and Future Directions:**

We also discussed the broader implications of our findings for sustainable waste management and resource recovery in the Result part (listed below). Our approach can **be adapted for other gaseous products involved electrochemical reactions**, potentially transforming waste treatment practices globally.

Line 389-398, page 24

Future research should include the electrosynthesis and separation of a more diverse array of bulk and fine chemicals. **Additionally, future studies should explore integrating pre-concentration processes for low-concentration waste streams and optimizing reactor or catalyst layer designs, such as zero-gap electrolyzers, flow-through electrodes, or coupled porous adsorption materials to overcome mass transfer limitations. Electrosynthesis based on waste streams presents a cost-effective alternative to traditional waste removal processes, maximizing the value extracted from complex, abundant wastewater resources. Its on-site deployment at wastewater treatment facilities or pollution sources supports the circular economy, promotes energy sustainability, and enables zero liquid discharge. This approach offers substantial economic, environmental, and societal advantages.**

2. The authors need to clarify in which aspect (conversion efficiency, recovery efficiency, or overall cost) the proposed system outperforms existing studies in terms of ammonium sulfate or sodium hypochlorite synthesis. The nitrate part can be relatively easily compared to existing studies as previously mentioned. The chloride part is concerning as the faradic efficiency is low (oxygen evolution is dominant), and how does it compare to typical Chlor-alkali process.

Specifically for brines, the sodium sulfate or sodium chloride can be crystallized and recovered in high purity. Further use of these sodium chloride for electrolysis to generate sodium hypochlorite seems more efficient and feasible.

Reply: Thank you for your insightful comments and for raising pertinent questions regarding the performance metrics of our electrolyzer system in comparison to existing studies on ammonium sulfate or sodium hypochlorite synthesis. Your observations have prompted a detailed re-evaluation of our system's capabilities, particularly in relation to the traditional Chlor-alkali process and the treatment of brines.

As highlighted in our manuscript, our system is specifically designed for the paired electrolysis of co-existing NO_3^- and Cl^- ions in wastewater, which are converted into NH_3 and Cl_2 (listed below). **This integrated approach underscores the significance of considering the system as a whole rather than isolating the cathodic and anodic processes for independent evaluation.** We appreciate this opportunity to clarify these aspects and provide a comprehensive response:

Line 54-68, page 4:

An important example is conversion of NO_3^- and Cl^- ions to NH_3 and Cl_2 gases, which are chemicals produced globally at approximately 182 million and 88 million metric tons per year, respectively.¹⁶⁻²⁰ NO_3^- and Cl^- are commonly present in industrial wastewater such as ion-exchange brines, which may contain 150 mM NO_3^- and 5 wt% NaCl.^{21, 22} Electrocatalytic conversion of nitrate to ammonia, which has been demonstrated,^{14, 23} involves cathodic nitrate reduction (Equation 1) coupled to an anodic reaction such as water oxidation (Equation 2).^{9, 24} Similarly, industrial chlorine gas (Cl_2) is primarily produced by the chlor-alkali process, which consists of an anodic chlorine-evolution reaction paired with a hydrogen (H_2)-evolution reaction (Equations 3 and 4).^{25, 26}

While today's processes separately generate O₂ and H₂, it makes sense to couple the nitrate-to-ammonia conversion with chlorine evolution.

Comparison of Performance Parameters:

- NO₃⁻ to NH₃ Conversion and Separation: Referencing Table 1, our NH₃ recovery efficiency surpasses those reported in other studies. Crucially, **our process is the only one where the market value of the obtained products exceeds the cost of electricity used**, contrasting with other studies where electricity costs outweigh profit. This underscores the benefits of our approach over traditional single pollutant treatment and resource recovery (such as NO₃⁻) which is hard to yield a net positive economic return.

Table 1. Comparison of performance parameters.

Papers:	NO ₃ ⁻ concentration in the feeding real wastewater (mM)	NH ₃ recovery efficiency (%)	Profit (market price of value-added chemicals minus electricity costs)
Environ. Sci. Technol. 2024, 58, 16, 7208–7216	17.6	99.3%	–\$5100 per metric ton of N
Water Research, 2023, 242, 120256	7.1	87.5%	–\$5700 per metric ton of N
Environ. Sci. Technol. 2021, 55, 15, 10684–10694	14.8	98.7%	–\$25417 per metric ton of N
This study	NO ₃ ⁻ : 11.8 Cl ⁻ : 54.1	NH ₃ : 99.6% Cl ₂ : ~100%	+\$2364 per metric ton of products ((NH ₄) ₂ SO ₄ and NaClO)

Cl₂ Synthesis: Regarding Cl₂ production, comparisons with other studies utilizing high-concentration synthetic NaCl electrolytes (over 1 M NaCl) for chlorine electrolysis or Chlor-alkali processes are inequitable. Because that high Cl⁻ concentrations are

uncommon in typical waste streams. Our use of 10-100 mM Cl^- more accurately reflects real-world conditions, making direct comparisons of Faradaic efficiencies somewhat misleading. While our chloride evolution reaction's Faradaic efficiency is relatively low, the overall process must consider the economic viability and environmental benefits of simultaneously synthesizing NH_3 and Cl_2 . Besides, as far as we know, this study is the first demo for the electrosynthesis and separation of Cl_2 .

- **Chlor-alkali process:** From 1888 till now, three processes emerged in the chlor-alkali industry: the diaphragm cell, mercury cell and present membrane cell.^{1,2}

For the industrial chlor-alkali process, in the diaphragm and membrane process, **the use of asbestos or fluorine-containing materials** causes indirect emissions. Because of these materials' limited lifetime (approximately several years), the diaphragm and membrane process results in **moderate environmental risks**.³

Furthermore, the membrane cell has dominated the current chlor-alkali industry. However, the general applications of this membrane-based chlor-alkali process remain challenging. Although this method can facilitate the separation of the products because of its chemical resistance, **the expensive ion exchange membrane generally exhibits limited useful life**.^{4,5}

The old mercury cell can be considered as a typical membrane-free chlor-alkali technology. Unfortunately, **because of the high toxicity, the mercury cell-based chlor-alkali technology must be stopped step by step**.⁶

Therefore, developing an environmentally friendly, high-efficiency, membrane-free chlor-alkali process is highly desired. **The membrane-free electrolyzer from waste streams, which could be deployed in wastewater treatment facilities or pollution sources, could be a supplement and/or new alternative for Cl_2 production.**

References cited in response letter:

1. O'Brien, Thomas F., Tilak V. Bommaraju, and Fumio Hine. Handbook of Chlor-Alkali Technology: Volume I: Fundamentals, Volume II: Brine Treatment and Cell Operation, Volume III: Facility Design and Product Handling, Volume IV: Plant Commissioning and Support Systems, Volume V: Corrosion, Environmental Issues, and Future Development. Springer US, 2005.

2. Lakshmanan, Shyam, and Thanapalan Murugesan. "The chlor-alkali process: work in progress." *Clean Technologies and Environmental Policy* 16 (2014): 225-234.
3. Moussallem, Imad, et al. "Chlor-alkali electrolysis with oxygen depolarized cathodes: history, present status and future prospects." *Journal of Applied Electrochemistry* 38 (2008): 1177-1194.
4. Madaeni, S. S., and V. Kazemi. "Treatment of saturated brine in chlor-alkali process using membranes." *Separation and Purification Technology* 61.1 (2008): 68-74.
5. Hine, F., et al. "Effect of silicate on the membrane-type chlor-alkali cell." *Journal of applied electrochemistry* 21 (1991): 781-784.
6. Hou, Mengyan, et al. "A clean and membrane-free chlor-alkali process with decoupled Cl₂ and H₂/NaOH production." *Nature communications* 9.1 (2018): 438.

- **Process Comparison:** Your suggestion to split the co-production of NH₃ and Cl₂ into two separate processes: (1) converting NO₃⁻ to NH₃ and separating NH₃, and (2) crystallizing sodium sulfate or sodium chloride from brines for subsequent NaClO production, which apparently overlooks the synergistic benefits of our integrated approach. As detailed in the Introduction (Lines 54-68, Page 4, listed above), separating these processes leads to the generation of less economically valuable by-products like H₂ and O₂, which diminish the overall economic advantage. In contrast, our integrated co-electrosynthesis of NH₃ and Cl₂ has demonstrated not only clear economic potential but also enhanced operational efficiency, reinforcing the value of our unified approach.

To address your concerns, we have revised the “conclusion” part to more emphatically highlight the advancements and broader applicability of our electrolyzer system. This revision underscores the significant improvements made and ensures the system's relevance across various applications.

Line 389-398, page 24

Future research should include the electrosynthesis and separation of a more diverse array of bulk and fine chemicals. Additionally, future studies should explore integrating pre-concentration processes for low-concentration waste streams and optimizing reactor or catalyst layer designs, such as zero-gap electrolyzers, flow-through electrodes, or coupled porous adsorption materials to overcome mass transfer limitations. Electrosynthesis based on waste streams presents a cost-effective alternative to traditional waste removal processes, maximizing the value extracted from complex, abundant wastewater resources. Its on-site deployment at

wastewater treatment facilities or pollution sources supports the circular economy, promotes energy sustainability, and enables zero liquid discharge. This approach offers substantial economic, environmental, and societal advantages.

Response to Reviewer #2

After reviewing the revisions and the responses provided by the authors, I am pleased to recommend acceptance of this manuscript. The authors have addressed the concerns raised during the review process thoughtfully and comprehensively, enhancing both the clarity and depth of the paper. The improvements made significantly contribute to the robustness and scientific merit of the work. This manuscript now presents a valuable contribution to the field and should be well-received by the community. I look forward to seeing this work published in Nature Communication.

Reply: Thank you for your encouraging feedback and for recommending the acceptance of our manuscript for publication in Nature Communications. We appreciate your acknowledgment of the enhancements made to our manuscript and are delighted to hear that our work now stands as a valuable contribution to the field. Your supportive words are greatly encouraging. Thank you once again for your thoughtful and constructive review process.

Response to Reviewer #3

My topics have been addressed, thank you. No further comments.

Reply: Thank you for your feedback and for confirming that all your concerns have been addressed satisfactorily. We greatly appreciate the time and effort you invested in reviewing our manuscript and for your constructive comments, which have undoubtedly enhanced the quality of our work.

Response to Reviewer #4

The authors did a good job to answer all the questions raised by the referees and I have no doubt that this manuscript can become feasible for publication in Nature Commun. Yet, I do have some additional questions that come up during reading the revised manuscript as well as the comments that need to be answered.

Reply: Thank you for acknowledging the revisions made to our manuscript and for considering it feasible for publication in Nature Communications. The responses to your additional questions have been answered one by one as listed below.

1. The authors write: "We acknowledge your point about extending the reaction time; however, extending beyond 5 hours tends to lead to increased product loss, which may not be reflective of practical electrosynthesis and separation scenarios. Hence, the decision to limit the duration to 5 hours was to maintain relevance to typical operational times."

Exactly this observation is a critical point that should be reported as it has significant impact on the process conditions and subsequent optimization strategies. Therefore, this behavior should be shown and explained. The authors should really show the performance beyond the 5 hours.

Reply: We agree and extend the electrolysis time from 5 h to 10 h. The data were updated in Fig 3 b-f. The relevant discussion has been updated to the section of "Probing the mechanism of NH₃/Cl₂ separation on reducing product loss".

Line 198-252, page 13:

To gain insights into the effect of NH₃/Cl₂ separation on reducing product loss, we conducted a series of control experiments by varying the concentrations/ratios of nitrogen and chloride species in the feed electrolyte, with and without the incorporation of separation operations. The major heterogeneous and homogeneous redox reactions within the electrolyzer are shown in **Fig. 3a** and summarized in Supplementary Table 1.³⁸ As shown in **Fig. 3b** and **c**, the introduction of Cl⁻ ions resulted in an observable increase in the remaining NO₃⁻ concentration (blue data points), from 1.4±0.3 mM to 9.0±0.3 mM, and a corresponding

decrease in the final NH_3 (purple data points) concentration, from 13.1 ± 0.2 mM to 1.7 ± 0.2 mM. The concentration of N_2 (green data points), encompassing both dissolved and vaporized forms, increased from 9.6 ± 0.3 mM to 14.3 ± 0.1 mM. This calculation was based on the disparity between the input nitrate nitrogen and the nitrogen species retained in the solution. After 4 hours, the stability of NH_3 concentration, despite a decreasing NO_3^- concentration, is attributed to the direct oxidation of NH_3 under alkaline conditions,³⁸ as indicated by the electrolyte pH rising above 11.5. The amplified N_2 concentration when introducing Cl^- is attributed to the more rapid reaction kinetics towards N_2 ($4.2\times 10^6 \text{ M}^{-1}\cdot\text{s}^{-1}$) compared to NO_3^- ($0.1\text{--}0.7 \text{ M}^{-1}\cdot\text{s}^{-1}$) in the context of $\text{NH}_3/\text{NH}_4^+$ interaction with HClO/ClO^- .^{27, 38} As the rate-limiting species for NO_3^- reduction, the average NO_2^- concentration within 10 h electrolysis (gray data points) reduced from 1.9 ± 0.1 mM to 0.9 ± 0.1 mM, when Cl^- was present.

Subsequent experiments aimed to trace the NO_2^- conversion pathways (e.g., conversion to NO_3^- or NH_3) and to ascertain the oxidation priorities of active chlorine species with NO_2^- and NH_3 . To this end, equal amounts of NO_2^- and NH_3 were introduced together to the electrolyte, and the evolution of subsequent nitrogen species was monitored. **Fig. 3d** indicates that, during the initial 3 h, the NO_2^- concentration decreased from 12.5 mM to 1.2 ± 0.2 mM and was mainly oxidized to NO_3^- that increased from 0 mM to 8.3 ± 0.2 mM. Meanwhile, the N_2 and NH_3 concentrations increased from 0 mM and 12.5 mM to 1.1 ± 0.2 mM and 14.4 ± 0.5 mM, respectively. After 3-hour when NO_2^- was nearly depleted, NH_3 oxidation became dominant as indicated by the reduced NH_3 concentration from 14.4 ± 0.4 mM to 2.2 ± 0.2 mM and the increased N_2 concentration from 1.1 ± 0.2 mM to 13.4 ± 0.4 mM. From a reaction kinetic standpoint, NH_3 has multi-step conversions with rate constants spanning from 1.7×10^2 to $3.1\times 10^6 \text{ M}^{-1}\cdot\text{s}^{-1}$ and is considerably more vulnerable to HOCl induced oxidation than NO_2^- that has multi-step conversions with measured rate constant for only one step by far ($1.8\times 10^5 \text{ M}^{-1}\cdot\text{s}^{-1}$).³⁹⁻⁴² The preferential reaction of NO_2^- with HOCl can be explained by the breakpoint chlorination mechanism in NH_3 , which occurs when the HOCl to NH_3 mole ratio gradually reaches 1.5.³⁸ The continuous consumption of HOCl by NO_2^- prevents the system from reaching this breakpoint chlorination ratio. During HOCl -mediated NH_3 oxidation to N_2 and NO_3^- , chloramine intermediates react with NO_2^- , converting back to NH_3 as end products.⁴² This process further influences the dynamics of NH_3/HOCl interactions. The main interference

arises from HOCl that converts NO_2^- to NO_3^- at a kinetic rate five orders of magnitude faster than chloramine reactions due to the low HOCl concentration condition within the electrolyzer.⁴³

We further assessed the influence of the separation of NH_3 and Cl_2 from the electrolyte on the final product formation. Attaching an ammonia trap channel next to the gas-permeable cathode (**Fig. 3e**) separated 99% of the produced NH_3 from the electrolyte channel (red data points). Consequently, the NO_3^- -conversion efficiency increased by 11%, and nitrogen loss evidenced by N_2 formation also decreased by 76%. A similar experiment was conducted by incorporating a single chlorine trap channel (without the use of the ammonia trap channel), which yielded a modest improvement of the NO_3^- conversion efficiency by 46% and the reduced N_2 loss by 25% according to the results in **Fig. 3f**. Although ~100% of the generated Cl_2 was extracted from electrolyte, after 5 h electrolysis, the Cl_2 concentration in chlorine trap solution (7.5 ± 0.5 mM) was still lower than that in **Fig. 2c** (9.1 ± 0.1 mM), where the separation of NH_3 and Cl_2 occurred simultaneously. This observation implies that the Cl_2 recovery efficiency of >98% across a broad anodic potential range in **Fig 1e** may be misleading. The Cl_2 extraction kinetics rate is not fast enough to efficiently separate all produced Cl_2 at the anodic interface, which consequently caused the residual HClO and ClO^- in the waste stream. These residual species are likely to engage in side reactions with NH_3 and NO_2^- , resulting in the formation of Cl^- . Therefore, the optimal electrosynthesis and separation performance of the membrane-free electrolyzer required synchronous extraction processes for NH_3 and Cl_2 . Isolating ammonia or chlorine gas alone was inadequate to mitigate products loss.

Fig. 3 | Mechanism analysis. b-f, the nitrogen species and active chlorine species evolution over reaction time when the electrolytes in the reaction chamber contained (b) 25 mM NO₃⁻; (c) 25 mM NO₃⁻, 25 mM Cl⁻; (d) 12.5 mM NO₂⁻, 12.5 mM NH₃, 25 mM Cl⁻, pH 12; (e) 25 mM NO₃⁻, 25 mM Cl⁻; (f) 25 mM NO₃⁻, 25 mM Cl⁻. All experiments were carried out under a cell potential of 3.0 V. The shadow area in the figure represents the error scale.

2. The new I-t curve presented in Figure S7 reveals a small but noticeable decay over time. The authors should explain why this is the case and how it could potentially be solved.

Reply: The current decay over time might be attributed to the decreased reactants (NO_3^- and Cl^- ions) concentration during electrolysis. We supplemented a set of I-t curve experiment (shown below) and detected the concentrations of NO_3^- and Cl^- in electrolyte at different time points to confirm this speculation.

Figure. I-t curve and corresponding NO_3^- and Cl^- concentrations.

3. In their conclusion, the authors provide a clarification on how to achieve higher current densities and argue that this is caused by mass transfer limitations under low waste concentrations. While this is partially true, the authors need to describe how to circumvent this problem since otherwise referee 1 has a valid point on the low novelty and the reactor itself renders useless for applications. Typically, the separation from mass transfer and electron transport is solved in membrane based systems by using zero-gap electrolyzers. How do the authors anticipate to make their system broadly applicable?

Reply: We greatly appreciate this insightful suggestion. We have modified the conclusion part to strength the electrolyzer improvement part and make the system broadly applicable.

Line 389-398, page 24:

Future research should include the electrosynthesis and separation of a more diverse array of bulk and fine chemicals. Additionally, future studies should explore integrating pre-concentration processes for low-concentration waste streams and optimizing reactor or catalyst layer designs, such as zero-gap electrolyzers, flow-through electrodes, or coupled porous adsorption materials to overcome mass transfer limitations. Electrosynthesis based on waste streams presents a cost-effective alternative to traditional waste removal processes, maximizing the value extracted from complex, abundant wastewater resources. Its on-site deployment at wastewater treatment facilities or pollution sources supports the circular economy, promotes energy sustainability, and enables zero liquid discharge. This approach offers substantial economic, environmental, and societal advantages.

REVIEWERS' COMMENTS

Reviewer #1 (Remarks to the Author):

The authors compared the differences between current manuscript and previous similar studies (Environ. Sci. Technol. 2024, 58, 16, 7208–7216; Water Research, 2023, 242, 120256; Environ. Sci. Technol. 2021, 55, 15, 10684–10694). While the concepts are similar, the authors did mention system optimization by integrating ammonia and chlorine trapping using more compact gas diffusion electrode design. However, this simple integration doesn't demonstrate significant scientific breakthrough, and is too incremental in science for high impact journals such as Nature Communications. Therefore, I still think there is a lack of novelty, but I respect the editor and other reviewers' opinions.

Reviewer #4 (Remarks to the Author):

The authors successfully addressed all the corrections I requested. While referee 1's concerns about novelty are valid, novelty is mostly a blend of objective data and subjective judgment. The readiness of the application and whether the combination of Cl₂ formation and NH₃ oxidation from waste streams constitutes something new is indeed debatable. However, the manuscript introduces some fresh perspectives. Despite any scientific reservations, my questions have been satisfactorily answered, and I am comfortable with the science. From a science based perspective, I do not have any objections towards its publication now.